# The cerebellum regulates fear extinction through thalamo-prefrontal cortex interactions in male mice

Jimena L. Frontera [1,2], Romain W. Sala[1,2], Ioana A. Georgescu[1], Hind Baba Aissa[1], Marion N. d'Almeida[1], Daniela Popa [1,3] & Clément Léna [1,3] ✉

Fear extinction is a form of inhibitory learning that suppresses the expression of aversive memories and plays a key role in the recovery of anxiety and trauma-related disorders. Here, using male mice, we identify a cerebello-thalamo-cortical pathway regulating fear extinction. The cerebellar fastigial nucleus (FN) projects to the lateral subregion of the mediodorsal thalamic nucleus (MD), which is reciprocally connected with the dorsomedial prefrontal cortex (dmPFC). The inhibition of FN inputs to MD in male mice impairs fear extinction in animals with high fear responses and increases the bursting of MD neurons, a firing pattern known to prevent extinction learning. Indeed, this MD bursting is followed by high levels of the dmPFC 4 Hz oscillations causally associated with fear responses during fear extinction, and the inhibition of FN-MD neurons increases the coherence of MD bursts and oscillations with dmPFC 4 Hz oscillations. Overall, these findings reveal a regulation of fear-related thalamo-cortical dynamics by the cerebellum and its contribution to fear extinction.

Impaired emotion regulation is a growing concern in modern societies, and is responsible for major behavioral dysfunctions. Indeed, the failure to suppress fear responses underlies several anxiety disorders, such as the post-traumatic disorder (PTSD). The extinction of conditioned fear has been an essential paradigm to identify the brain networks and neural mechanisms involved in the suppression of fear[1,2]. Growing evidence indicates that the cerebellum has multiple connections with the fear network[3–5] review in ref. [6], and exhibits BOLD activations during emotion processing, notably in the medial part of the cerebellum[7–9]. Recent evidence showed that the cerebellum is also involved in fear extinction[10–12] and exhibits anomalous functional connectivity with the emotional network in PTSD[13,14]. However, little is known on how this structure regulates fear memories.

Fear extinction results from the formation of a new memory trace, and is known to heavily rely on the medial region of the prefrontal cortex (mPFC), composed of the dorsomedial (dmPFC) and ventromedial (vmPFC) subdivisions, which play complementary roles in fear extinction[15–18]. The mPFC is closely linked to other limbic structures (eg. amygdala, hippocampus)[19,20], and is reciprocally connected to the mediodorsal thalamic nucleus (MD)[21–24], known to have a substantial contribution in fear extinction learning[25–29]. Strikingly, dual firing patterns have been associated with extinction learning in MD thalamo-cortical neurons[27]. While the increase of tonic firing in MD facilitates fear extinction, burst firing in MD neurons prevents fear extinction learning.

Moreover, the interactions between the different structures of the limbic network in the course of emotional processing have been associated with neuronal synchronization[30–32]. Notably, the expression of conditioned fear memories has been associated with prominent synchronous 4 Hz oscillations within dmPFC-basolateral amygdala (BLA) circuit, which organize the spiking activity of local neuronal populations and cause fear memory expression[30,33,34].

[1]Neurophysiology of Brain Circuits Team, Institut de Biologie de l'Ecole Normale Supérieure (IBENS), Ecole Normale Supérieure, CNRS, INSERM, PSL Research University, 75005 Paris, France. [2]These authors contributed equally: Jimena L. Frontera, Romain W. Sala. [3]These authors jointly supervised this work: Daniela Popa, Clément Léna. ✉e-mail: clement.lena@bio.ens.psl.eu

While the contributions of the mPFC, BLA, and MD to fear extinction have been studied in detail, much less is known about the contribution of the cerebellum. The cerebellum has been associated with fear learning and freezing behavior[35,36], review in ref. [37]. Associative fear learning triggers the long-term potentiation of synapses in the cerebellar cortex[38], and cerebellar vermis inactivation during consolidation phase weakens fear-related memories[39], suggesting that the cerebellar plasticity participates to fear processing (but see ref. [40]). Recently, we have described a role of the cerebellum in fear memories through its projections to the ventrolateral periaqueductal grey (vlPAG)[3]. However, little is known about the interconnections of the cerebellum with other limbic-related areas and their contributions to the neurophysiological mechanisms involved in fear extinction.

In this study, we address these questions with a combination of neuroanatomical tracing, chemogenetic manipulations, optogenetics and electrophysiology in freely moving mice, and we describe a pathway that links the cerebellum with the dmPFC via the MD and regulates fear extinction.

## Results

### The MD nucleus is a thalamic relay between the FN and the dmPFC

In order to examine the connectivity between the cerebellum and the thalamic MD nucleus, known to play an important role in fear extinction, we injected a retrograde adeno-associated virus (AAV) in MD to induce the expression of GFP in neurons projecting to this area (Fig. 1a). We found in the cerebellum a robust retrograde labeling in the cerebellar fastigial nucleus (FN) (Fig. 1b), notably in the caudal part (Fig. 1c), and more abundantly in the medial area of the FN (Fig. 1d). The distribution of retrogradely labeled neurons in FN were highly

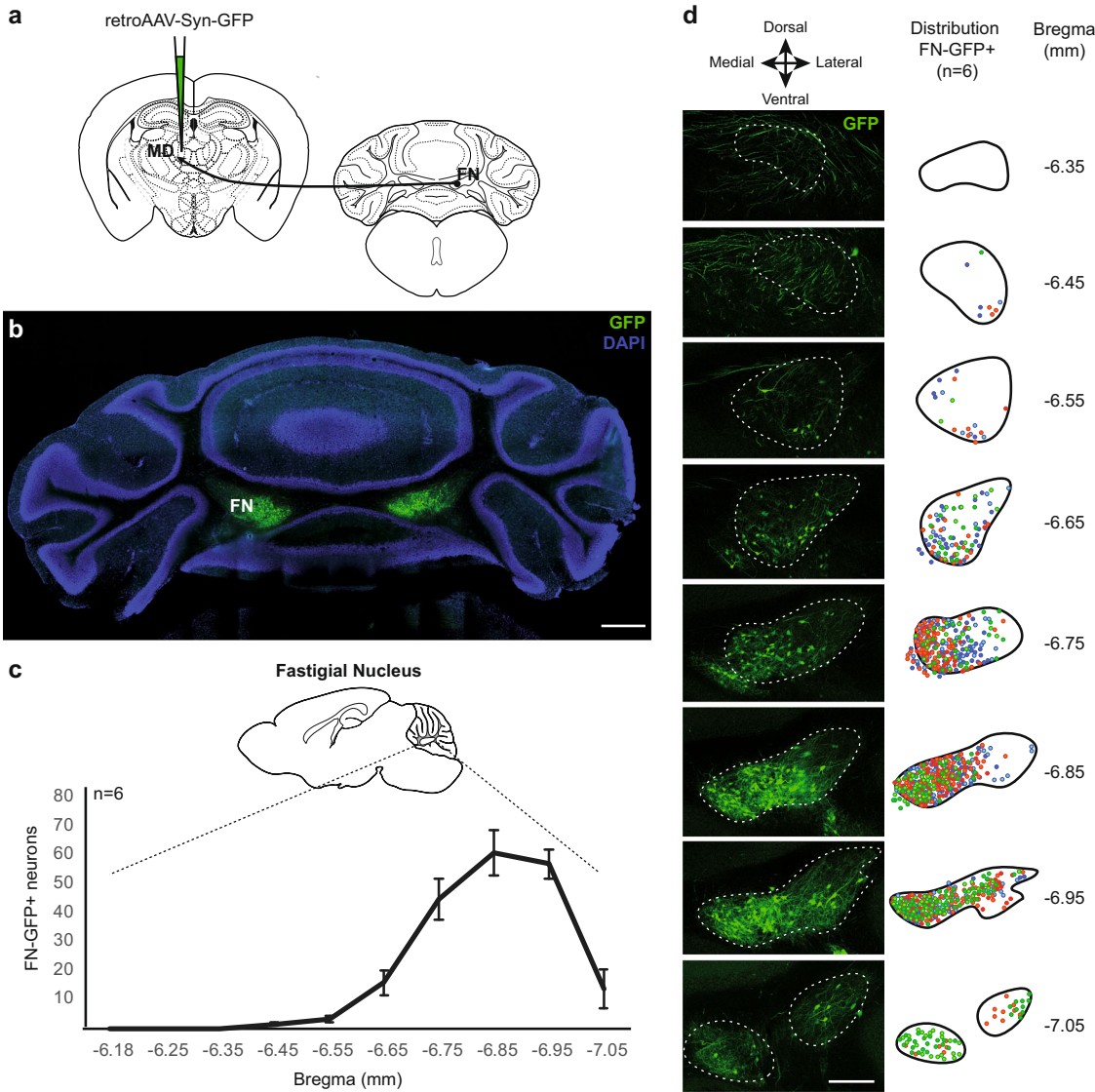

**Fig. 1 | Cerebellar fastigial nucleus (FN) sends projections to the mediodorsal thalamic nucleus (MD). a** Retrograde tracing strategy by injection of retrograde AAV-GFP in the MD. **b** Coronal cerebellar sections of FN with MD-projecting neurons expressing retrograde AAV-GFP (green), and cell nuclei labeled with DAPI (scale bar, 500 μm). **c** Quantification (mean +/− SEM) and antero-posterior distribution of retrograde GFP-labeled FN MD-projecting neurons (FN-GFP+) in coronal sections (data from 6 mice). AP position is reported relative to Bregma. **d** Distribution pattern of FN MD-projecting neurons (FN-GFP+) in coronal cerebellar sections across replicates (*n* = 6 mice, colors correspond to labeling from different animals). (scale bar, 250 μm). See also Supplementary Fig. 1. Brain schematic in panel **a** modified from the Allen Mouse Brain Atlas and Allen Reference Atlas – Mouse Brain[73,74] http://atlas.brain-map.org/atlas?atlas=1#atlas=1&plate=100960384, http://atlas.brain-map.org/atlas?atlas=1#atlas=1&plate=100960136, http://atlas.brain-map.org/atlas?atlas=1#atlas=1&plate=100960240.

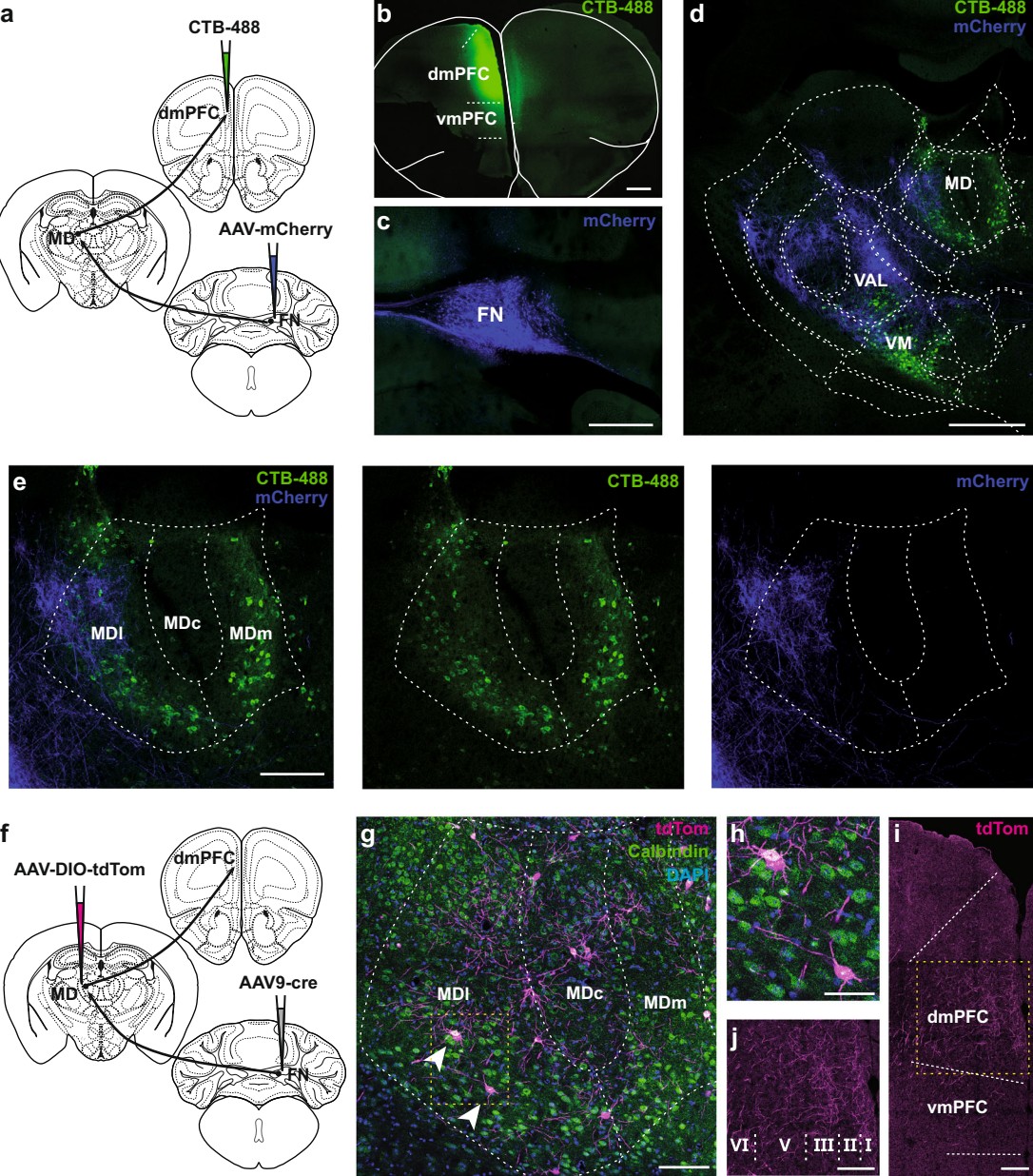

**Fig. 2 | MD as a high-order thalamic relay between the FN and the dmPFC.**
**a** Strategy for neuroanatomical tracing by injections of anterograde AAV-mCherry in the FN and retrograde CTB-488 in the dmPFC (*n* = 3 replicates). **b** Example of CTB-488 injection site in the dmPFC (scale bar, 500 μm). **c** AAV-mCherry expression in the cerebellar FN (scale bar, 500 μm). **d** Thalamic section exhibiting contralateral FN projections (mCherry, blue) and thalamic dmPFC-projecting neurons (CTB-488, green) (scale bar, 500 μm). **e** Zoom-in from thalamus section in **d** (dotted line area), showing the FN projections to MD, preferentially into the lateral segment of the MD (MDl). MDc: central segment of MD, MDm: medial segment of MD. (scale bar, 100 μm). **f** Schematic representation of viral strategy to localize FN-post-synaptic neurons in MD. Anterograde trans-synaptic expression of cre by AAV9 and cre-dependent expression of td-Tomato in MD. (*n* = 3 replicates) **g** MD exhibiting FN-

post-synaptic labeled neurons (arrow heads) with tdTom and co-localization of calbindin expression detected by immunostaining (green). Cell nuclei labeled with DAPI (blue). **h** Zoom-in from **g** (yellow dotted line area) exhibiting FN post-synaptic MD neurons. **i** dmPFC sections exhibiting FN-post-synaptic MD neuronal projections in different cortical layers (scale bar, 100 μm). **j** Zoom-in from dmPFC section in **i** (yellow dotted line area), cortical layers I-VI (scale bar, 100 μm). Brain schematics modified from the Allen Mouse Brain Atlas and Allen Reference Atlas – Mouse Brain[73,74] from http://atlas.brain-map.org/atlas?atlas=1#atlas=1&plate=100960136, http://atlas.brain-map.org/atlas?atlas=1#atlas=1&plate=100960240 in panels **a**, **f**, and from http://atlas.brain-map.org/atlas?atlas=1#atlas=1&plate=100960260 in panels **d**, **e**, **g**.

consistent across replicates (*n* = 6, Fig. 1d). Furthermore, we also found cortical retrograde-labeled neurons projecting to MD, localized principally in the dmPFC (composed by the anterior cingulate cortex (ACC) and the prelimbic area (PL)) and less in vmPFC (mainly infralimbic cortex (IL)) (Supplementary Fig. 1), consistent with previous studies showing mPFC projections to MD[21,41].

Since the MD is reciprocally connected with the dmPFC[21–24], we examined whether the areas of the MD that receive projections from

the cerebellar FN correspond to MD areas projecting to the dmPFC. Using a combination of anterograde expression of AAV-mCherry in the FN (Fig. 2a, c) and the retrograde Cholera-Toxin subunit B (CTB) injection in the dmPFC (Fig. 2a, b), we found that the MD neurons projecting to the dmPFC were preferentially localized in the medial and the lateral MD (Fig. 2d, e), while the FN projections were principally localized contralaterally in the lateral MD, surrounding the MD neurons that project to the dmPFC (Fig. 2d, e). Thus, these

results suggest the existence of a di-synaptic FN-MD-dmPFC pathway.

To confirm the existence of this pathway, we used a trans-synaptic strategy combining injections of anterograde trans-synaptic AAV serotype 9 encoding cre-recombinase (cre) in the FN, with injections of anterograde cre-dependent AAV-DIO-tdTomato in the MD (Fig. 2f). We found tdTom-expression in MD neurons as a result of the anterograde trans-synaptic transport of AAV9-cre from FN terminals (Fig. 2g). Moreover, since MD dmPFC-projecting neurons have been reported to express calbindin, we examined calbindin-immunoreactivity and found that those tdTom-labeled MD neurons expressed calbindin (Fig. 2g, h). Finally, we observed the tdTom-labeled projections of MD neurons in the Layers I, III, V and VI of the dmPFC (Fig. 2i, j), confirming that the MD neurons receiving FN inputs project to the dmPFC. Taken together, our findings reveal the existence of a cerebello-dmPFC pathway through the lateral part of MD that also receives inputs from the dmPFC.

## Optogenetic stimulations of FN induce short latency responses in the MD and dmPFC

To study whether the FN input can drive MD and dmPFC neuronal activity, we performed single-unit recordings in MD and dmPFC coupled with optogenetic stimulation of Channelrhodopsin-2 (ChR2)-expressing neurons in the FN in freely moving mice (Fig. 3a, b). Optogenetic stimulation (100 ms) of the FN induced an increase in firing rate during light pulses, both in MD and dmPFC (Fig. 3c), while FN illumination in absence of ChR2 expression did not induce variation of the firing rate at the population level (Supplementary Fig. 2, all statistics are detailed in the Supplementary File). During optogenetic stimulation of the FN, we found that 49% of the MD recorded cells (77/158 MD neurons, $n = 10$ mice) displayed a significant increase in firing, whereas in the dmPFC 47% of the population (17/36 dmPFC neurons, $n = 7$ mice) significantly increased their firing rate during the light stimulation. Furthermore, among the responsive neurons in the MD and the dmPFC, a smaller set (MD: 14 cells with latency <15 ms; dmPFC: 8 cells with latency <20 ms) displayed short latencies of response (Fig. 3d). While MD neurons showed an early and sustained response, dmPFC neurons exhibited an initial increase in firing rate that was further amplified around 50 ms of stimulation (Fig. 3e, f). These data suggest a potent excitatory effect of the FN input to MD and dmPFC involving direct and indirect pathways.

Then, to test the direct and specific contribution of FN input to MD neuronal responses, we combined retrograde AAV-cre injection in the MD with cre-dependent AAV-DIO-ChR2 injection in the FN (Fig. 3g, $n = 10$ mice). In these experiments, short illumination (10 ms) of the FN MD-projecting neurons expressing ChR2 produced short latency responses in MD neurons (9/90 with latency ≤ 15 ms, Fig. 3h). In contrast, no short latency responses were observed in the dmPFC but some neurons exhibited a late increase in discharge (9/48 with latencies >20 ms, Fig. 3h). Thus, these results indicate a monosynaptic excitatory connection between the FN and the MD, and in concordance with a modulatory role of the MD on dmPFC activity in vivo[42], the activation of this pathway did not elicit a potent short-latency activation of the dmPFC neurons.

## Chemogenetic inhibition of the FN input to MD impairs fear extinction

Since the MD and the dmPFC are involved in fear extinction[15–18,27,28], we then investigated the contribution of the FN-MD pathway to this learning. For this purpose, mice were subjected to a Pavlovian fear conditioning protocol (FC), which consisted of 5 conditioned stimulus (CS, tone)-unconditioned stimulus (US, electrical foot-shock) paired presentations, followed by three consecutive days of fear extinction sessions (EXT), 25 unreinforced CS presentations per session (Fig. 4d).

We examined the contribution of the FN input to MD during fear extinction by specific chemogenetic inhibition of these projections. We selectively expressed inhibitory DREADD receptors in the FN neurons that target the MD, by injecting bilaterally retrograde CAV2-cre in the MD, and cre-dependent AAV- hM4Di (named Gi) in the FN (Fig. 4a–c). We then performed the FC and extinction sessions in these mice (Fig. 4d) and quantified the fear response as the percentage of time freezing during the CS.

After fear learning, control DREADD-expressing mice (saline-injected, CT + SAL) extinguished the freezing response over 3 days of extinction sessions (EXT1-3) (Fig. 4e). By contrast, mice under FN-MD input inhibition (Gi+CNO) during the first two days of extinction (EXT1 and EXT2), exhibited an impairment of fear extinction (Fig. 4e, Supplementary Fig. 3a, e). Both groups of mice had similar freezing levels at the beginning of EXT1 (median CT + SAL = 68.1%, Gi + CNO = 71.2%). Comparing mice expressing high levels of freezing during early EXT1 (above median) to mice expressing low levels freezing (below median) revealed that the impairment of extinction was only visible in the Gi+ CNO mice with the highest initial EXT1 freezing levels (Fig. 4e, top), in which little if any extinction occurred during EXT1 and EXT2. In contrast Gi + CNO mice with lower initial EXT1 freezing levels exhibited normal extinction suggesting that FN MD-projecting neurons primarily modulate extinction of high fear responses. During EXT3, without chemogenetic inhibition, both groups of mice showed a decrease in conditioned-fear response compared to EXT1 and EXT2, suggesting that the inhibition of FN MD-projecting neurons partially suppressed the expression of the learned extinction. Moreover, the basal freezing levels during the habituation to context A (before FC) or context B (before EXT1, EXT2 or EXT3, after SAL/CNO administration), were not significantly different between groups (Fig. 4f), indicating that chemogenetic inhibition did not induce general freezing behavior, but affected specifically the extinction of the freezing response to the CS. We verified that the dose of CNO used i.p. in our experiments had no effect per se on freezing or extinction learning, by performing FC and EXT sessions in sham mice (mice subjected to the same surgery procedure with infusion of AAV-mCherry instead of DREADD), injected with saline or CNO during EXT1 and EXT2. No differences were found between CNO and saline sham control mice, indicating that CNO had no effect in the absence of DREADD expression (Supplementary Fig. 3b, f). Altogether, these results indicate that FN neurons projecting to the MD modulate fear extinction.

To investigate if the effect observed by inhibition of FN MD-projecting neurons involves the FN inputs to the MD, we induced local inhibition of the DREADD-expressing FN terminals in MD, by intracranial local infusion of CNO in MD. In this set of experiments, mice expressing Gi in the FN-MD terminals received either SAL or CNO (0.5 mM) infusions, while another group of sham mice received the CNO (0.5 mM) infusion, 10 min before the EXT1 and EXT2. In concordance with the effect observed following the systemic administration of CNO, freezing levels were significantly higher in the Gi-CNO group at late EXT1, compared to both CT + SAL and CT + CNO control groups (Supplementary Fig. 3c, g). This effect was not observed in EXT2, but this may be due to the experimental limitations of repeated intracranial infusions. Overall, this result supports the hypothesis that FN influence on fear extinction is mediated by its input to MD.

The increased freezing observed during extinction following FN-MD inhibition could result from a disruption of the expression of conditioned fear (potentiation of fear expression, or disruption of fear suppression) rather than an effect on the extinction learning processes. If this was the case, FN-MD inhibition only during the EXT3 should result in an increased freezing. However, this experiment did not reveal significant differences between the FN-MD inhibited and control group (Supplementary Fig. 3d, h), indicating that this pathway does not simply modulate fear expression but also the extinction learning.

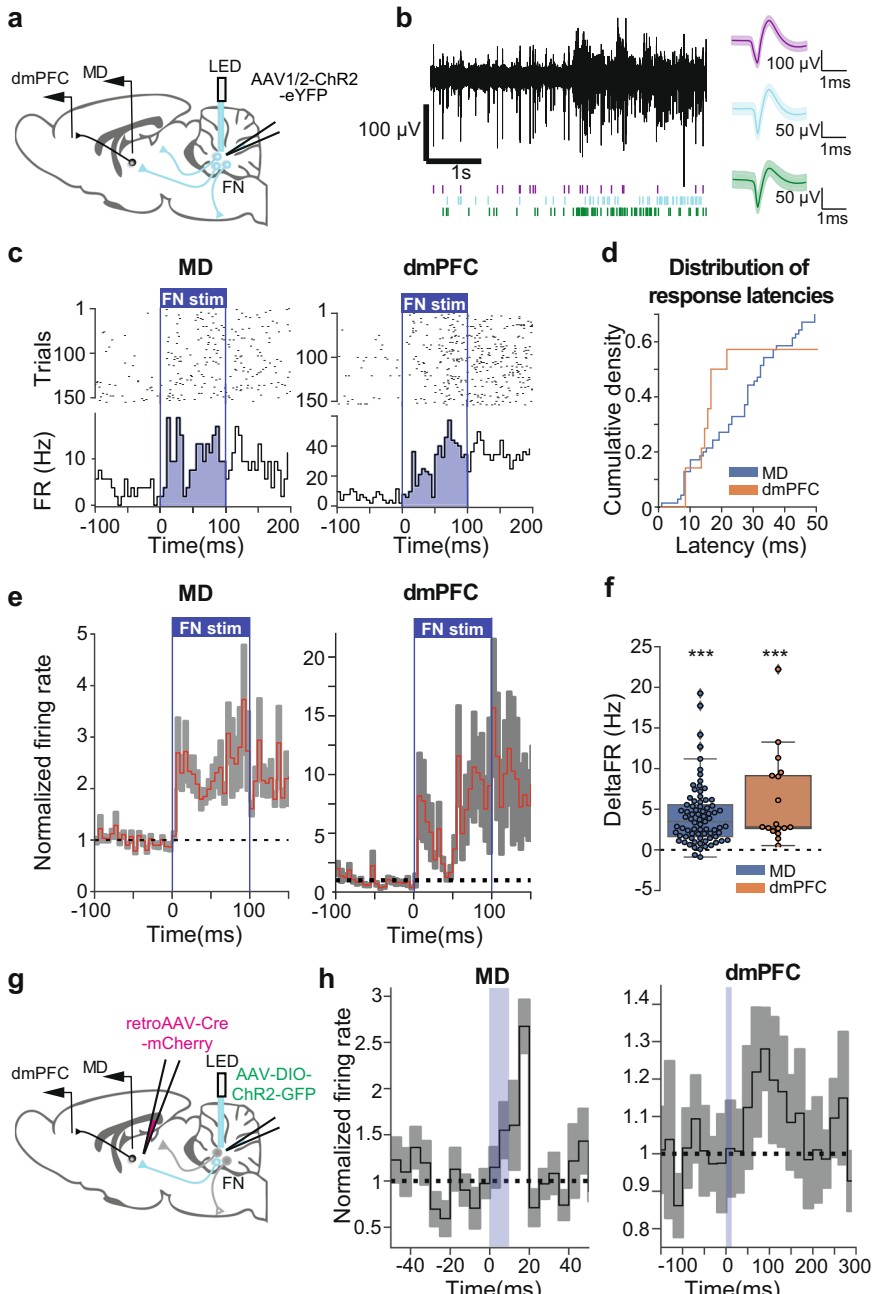

**Fig. 3 | Optogenetic stimulation of FN input to MD induced responses in MD and dmPFC. a** Strategy used for the specific optogenetic stimulation of FN neurons, representing the local injection of anterograde AAV-ChR2-eYFP in the FN and the implantation of recording electrodes in MD and dmPFC. **b** Example of high-passed filtered trace of a recording channel in MD (left) and spike shapes of single units from MD (average ± SD), after spike sorting (right). **c** Example PSTH (5 ms bins, bottom) and rasterplot (top) of a MD cell (left) and dmPFC cell (right) during 100 ms optogenetic stimulation of the FN. The light stimulation is represented by a blue rectangle. **d** Cumulative histograms showing the latency of neuronal response of responsive cells in MD and dmPFC triggered by 100 ms optogenetic stimulation of the FN. **e** PSTH (5 ms bins) displaying the change in firing rate (average ± SEM) of responsive cells during 100 ms optogenetic stimulation of the FN in MD (left), and in dmPFC (right). The light stimulation is represented by a blue rectangle. **f** Average

change in firing rate of responsive cells during 100 ms optogenetic stimulation of the FN. Wilcoxon test, (77 MD neurons from 10 mice, 17 dmPFC neurons from 7 mice) ***$p < 0.001$. Boxplots represent quartiles and whiskers correspond to range; points are singled as outliers if they deviate more than 1.5 x interquartile range from the nearest quartile. **g** Strategy used for the specific optogenetic stimulation of the FN inputs in MD, by injection of retrograde AAV-cre-mCherry in the MD and anterograde cre-dependent AAV-DIO-ChR2-GFP in the FN, and the implantation of recording electrodes in MD and dmPFC. **h** PSTH displaying the change in firing rate (average ± SEM) of responsive cells following 10 ms optogenetic stimulation of the FN in MD (left, 5 ms bin), and in dmPFC (right, dmPFC, 20 ms bin). Data available at doi:10.5061/dryad.9kd51c5ng. Detailed statistical results are available in the Supplementary Tables referenced by panel numbers.

Overall, these data indicate that the FN input to MD modulates the fear extinction in normal conditions, and that inhibiting this input leads to a deficiency in the extinction of the fear response.

## FN input to the MD is not involved in anxiety behavior or nociception sensitivity

The mPFC-limbic circuit is also recruited in anxiety[19]. Thus, in order to test whether the FN input to MD contributes to anxiety-like behavior, we

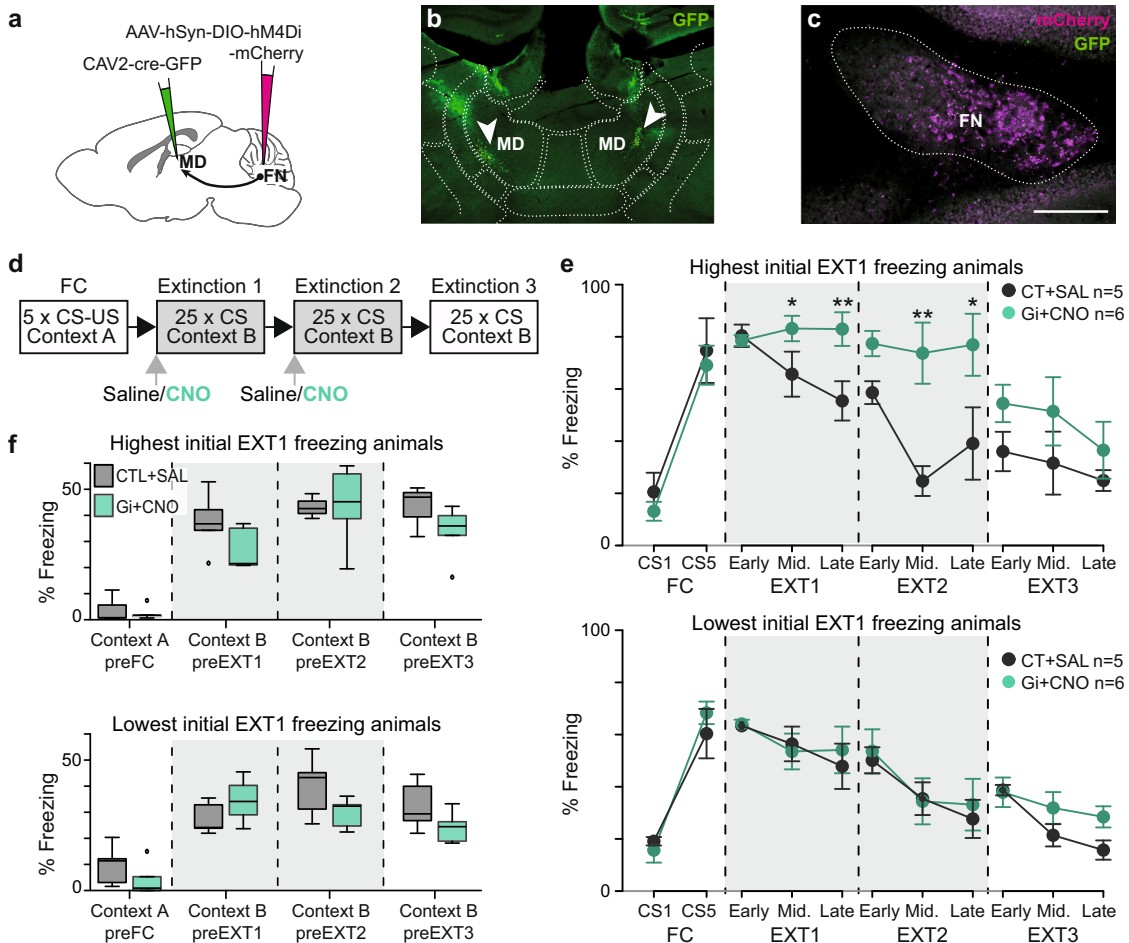

**Fig. 4 | FN input to MD modulates fear extinction. a** Chemogenetic strategy to inhibit specifically the activity of FN-MD input, bilateral retrograde expression of cre recombinase in MD by CAV2-cre-GFP infusion and anterograde cre-dependent expression of inhibitory DREADD (hM4Di (Gi)) in the cerebellar FN. **b** Injection site of CAV2-cre-GFP in the MD. **c** Cre-dependent expression of inhibitory DREADD reported by mCherry fluorescence in the FN (scale bar, 250 μm). **d** Classical fear conditioning and extinction protocol used. FN-MD projections were inhibited by CNO administration during extinction 1 and 2. **e** FN-MD input inhibition (Gi+CNO, $n = 12$) during extinction sessions 1 and 2 reduced extinction of fear response compared to the control group (CT + SAL, $n = 10$). The mice were separated in two groups of same size expressing respectively the highest and lowest freezing levels on the Early stage of EXT 1. Lines represent mean ± SEM. Post-hoc two-sided $t$-test Holm-Sidak corrected, $*p < 0.05$. **f** FN-MD chemogenetic inhibition of FN-MD input did not affect basal levels of freezing during habituation to the context of fear conditioning (Context A) or extinction (Context B), compared to the control mice; same groups and statistics as in panel **e**. Boxplots represent quartiles and whiskers correspond to range; points are singled as outliers if they deviate more than 1.5 x interquartile range from the nearest quartile, $p > 0.05$. Data available at doi:10.5061/dryad.9kd51c5ng. Detailed statistical results are available in the Supplementary Tables referenced by panel numbers.

performed different anxiety tests under chemogenetic inhibition of FN-MD projections: open field (Supplementary Fig. 4a), elevated plus maze (Supplementary Fig. 4b) and dark-light box (Supplementary Fig. 4c). There were no differences between the control and the FN-MD inhibition groups. Overall, these results indicate that the inhibition of FN MD-projecting neurons do not significantly affect anxiety behavior.

Since the MD also receives spinal nociceptive inputs and it is involved in the processing of negative affective aspects of pain[43], we examined the sensitivity of the mice to painful stimuli, using hot plate and tail immersion tests (Supplementary Fig. 4d). The results showed no alteration in the sensitivity under inhibition of FN input to MD, indicating that the inhibition of FN-MD projections does not modulate nociception.

Overall, our behavioral results suggest that cerebellar FN-MD projections contribute more specifically to the fear extinction learning process than to other MD-dmPFC functions related to negative emotions, such as anxiety and nociception.

### Inhibition of the FN input to MD increases thalamic bursting

Previous work showed that high-frequency bursts of action potentials in MD prevented fear extinction while an increased tonic firing in the MD promoted fear extinction learning[27]. Since inhibition of FN input to MD impaired fear extinction, an increased bursting in the MD could be a possible mechanism that explains our findings. Thus, we examined how the chemogenetic inhibition of the FN input to MD affects burst firing in the MD (Fig. 5). We used the Robust Gaussian Surprise method[44] (Fig. 5a, b) to detect bursts in the MD and quantify two bursting parameters: the burst occurrence (number of bursts per second) and firing rate within-bursts. All MD cells exhibited some bursting activity.

The inhibition of FN-MD input during EXT1 induced an increase in burst occurrence compared to the FC, both in the baseline (before first CS presentation) and during CS presentation (Fig. 5c, d). Notably, the burst occurrence in EXT1 was higher in the Gi + CNO group compared to the control, during baseline (when fear expression is enhanced, Fig. 4f right) and during CS presentation (Fig. 5c), and was reversed to values close to baseline during EXT3. While the burst occurrence was increased by the chemogenetic inhibition of FN-MD input, the firing rate within-burst was not different than the control mice, suggesting that bursting intensity in the MD was not altered by the disruption of FN-MD (Fig. 5d). Taken together, these data indicate that the FN-MD

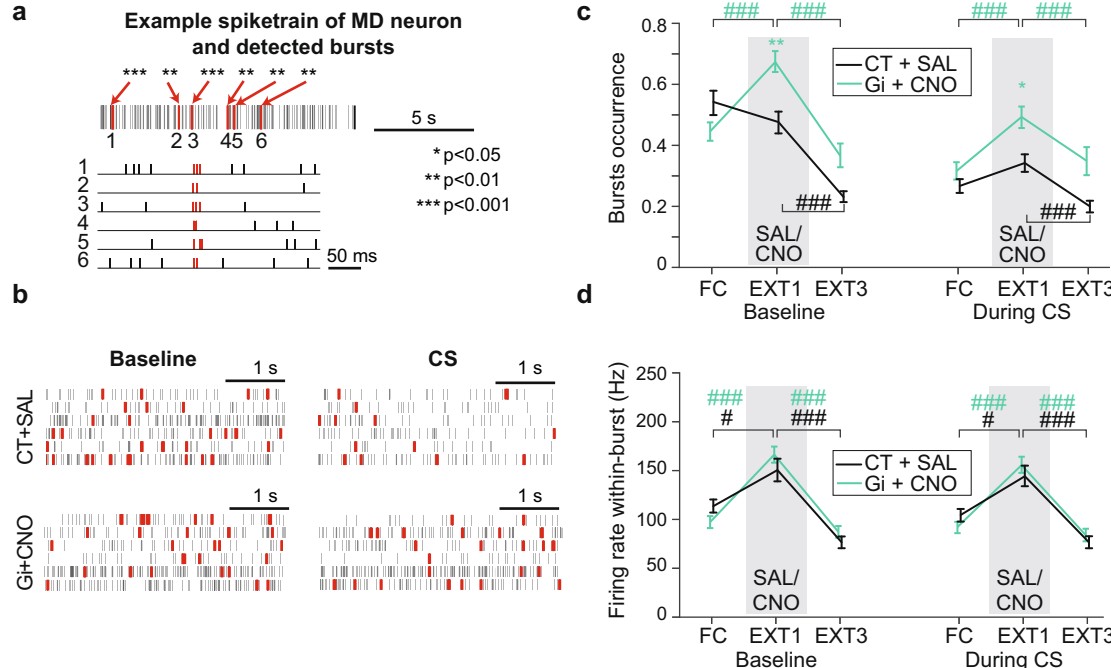

**Fig. 5 | Chemogenetic inhibition of FN-MD input during EXT1 increases burst occurrence in the MD. a** Example of burst detection in a spiketrain from a MD neuron, bursts are represented in red, and their associated *p*-values are reported. One sample *t*-test on ISI of bursts compared to the local distribution of ISI. *p < 0.05, **p < 0.01, ***p < 0.001. **b** Example spiketrains of MD neurons during Baseline (left), and during CS (right), from a CT + SAL mouse (top), Gi + CNO mouse (bottom). Each line corresponds to a given neuron, bursts are represented in red. **c** Burst occurrence (number of bursts per second) during baseline (left), and during CS (right). **d** The average firing rate within a burst during baseline (left) and during CS

(right). Comparisons between Gi + CNO (FC, EXT1, EXT3 *n* = 49, 67, 51 from 5 mice) and CT + SAL (*n* = 36, 39, 22 from 4 mice) for each phase are shown by colored stars on top of the corresponding phase Holm-Sidak corrected Mann–Whitney *U* test, *p < 0.05, **p < 0.01, ***p < 0.001. Comparisons between phases for each experimental group are shown by colored stars on top of line between two phases. Holm-Sidak corrected Mann–Whitney U test, #p < 0.05, ##p < 0.01, ###p < 0.001. Data available at doi:10.5061/dryad.9kd51c5ng. Data are ploted as mean +/− SEM. All tests are two-sided. Detailed statistical results are available in the Supplementary Tables referenced by panel numbers.

input participates in the regulation of burst firing in MD neurons and its inhibition results in an increased MD bursting activity in periods where fear expression is higher.

### dmPFC 4 Hz oscillations and neuronal dynamics in the MD

During fear extinction, the fear expression (conditioned-freezing) is associated with 4 Hz olfactory-entrained oscillations in the dmPFC and amygdala, anticipating and causing freezing occurrence[30,45]. To further investigate the interactions between MD and dmPFC during fear extinction, we studied how MD neuronal activity is related to the dmPFC 4 Hz local field potential (LFP) oscillations.

Consistently with the observations of Karalis et al.[30], we observed high levels of 4 Hz oscillations in the LFP of the dmPFC induced by the presentation of the CS in EXT1 (Fig. 6a), with peaks of Power Spectrum Density (PSD) in the 4 Hz range (2–6 Hz) (Fig. 6b). We found an increased fraction of the PSD associated to 4 Hz oscillations during CS compared to baseline in both experimental groups (Fig. 6c), consistent with the increase in fear responses during the CS. Moreover, the fraction of the PSD associated to 4 Hz oscillations during CS was increased under inhibition of FN input to MD, compared to control mice, consistent with the higher expression of fear response in this group.

Next, we examined the modulation of MD activity by dmPFC 4 Hz oscillations during CS presentations. MD neuronal activity exhibited a negative correlation with the amplitude of dmPFC 4 Hz oscillations during CS in both experimental groups, indicating a reduction of MD firing rate during high 4 Hz in most cells (Fig. 6d, e). We then identified episodes of high 4 Hz oscillations and found a significant reduction in burst occurrence in the MD during these episodes (Fig. 6f). However, the down-modulation of MD bursting by dmPFC 4 Hz oscillations was

significantly milder during CS when the FN input to MD was inhibited (Fig. 6f), yielding an overall higher burst occurrence during episodes of high dmPFC 4 Hz oscillations during CS (Fig. 6g).

All together, these data indicate an inhibition of the MD by dmPFC 4 Hz oscillations associated with freezing during fear extinction, and that the disruption of the FN-MD pathway during extinction results in an increased bursting during CS-related freezing.

### Phase-locking of MD bursting to dmPFC 4 Hz oscillations during fear extinction

Sustained fear expression relies on the maintenance of dmPFC 4 Hz oscillations[46]. We therefore examined how MD bursting is related to the dynamics of these oscillations. The average dmPFC LFP around MD bursts during episodes of high dmPFC 4 Hz revealed a slow oscillatory component centered on a positive peak in the dmPFC LFP, with a peak-to-peak latency compatible with a 4 Hz oscillation (Fig. 7a), suggesting that the MD bursting is organized around positive peaks of dmPFC 4 Hz oscillations. In agreement with this, the MD bursting (pooled across mice) displayed a significant phase-locking to dmPFC 4 Hz in both control and FN-MD inhibited mice, with a preferred phase close to the positive peaks of dmPFC 4 Hz oscillations (Fig. 7b), but the amplitude of modulation appeared larger following FN-MD inhibition. In concordance with this global MD observation, individual MD neurons showed phase-locking around a preferred phase close to the positive phase of dmPFC LFP (Fig. 7c), and a higher concentration of the individual preferred phases (measured by the distribution of angular distance between preferred bursting phases and dmPFC 4 Hz positive peak) was observed under FN-MD input inhibition indicating a more consistent phase locking of MD bursting to dmPFC 4 Hz oscillations in this condition (Fig. 7d, Supplementary Fig. 5a). These results

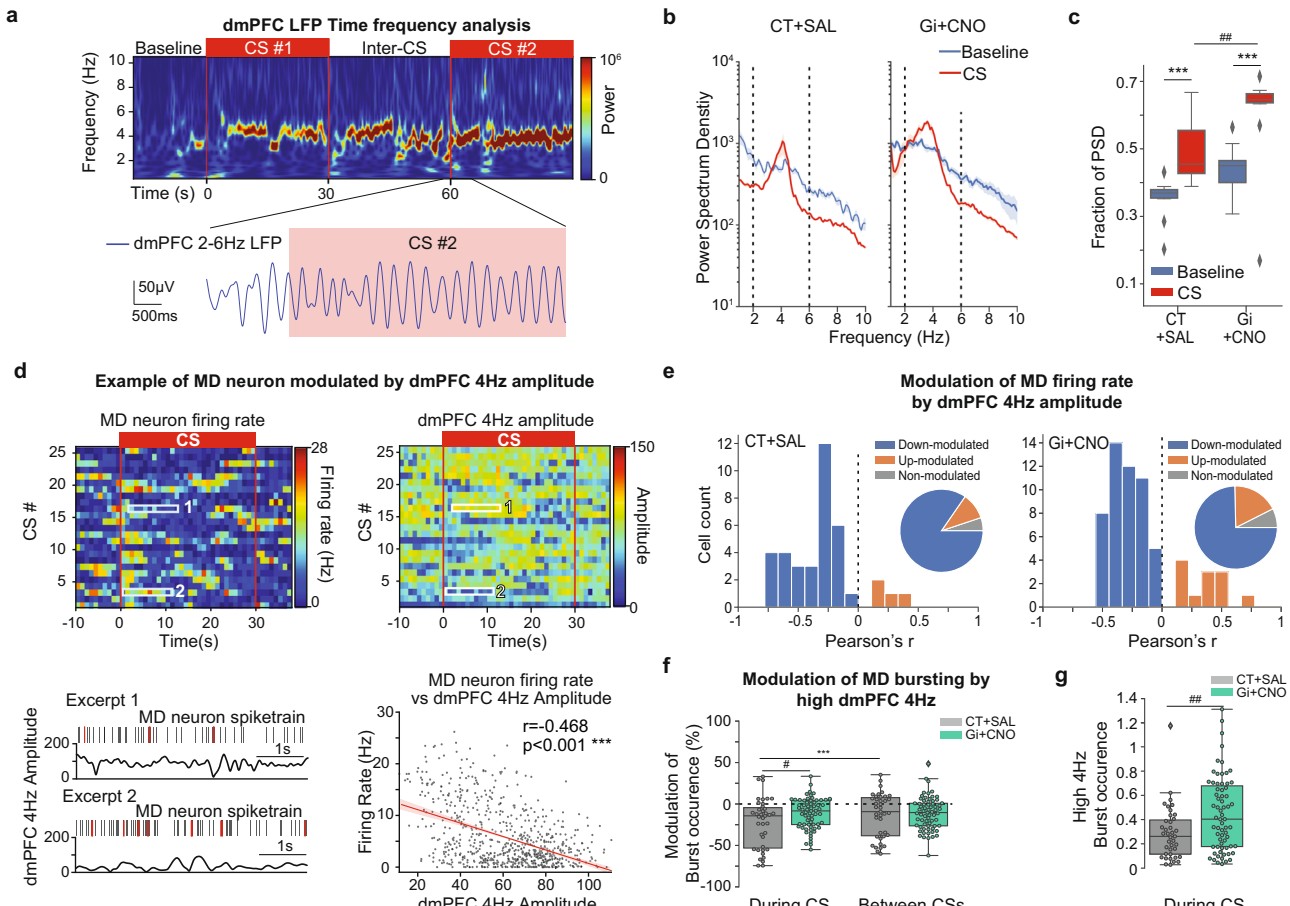

**Fig. 6 | Neuronal activity in the MD is modulated by dmPFC 4 Hz LFP oscillations during EXT1. a** Representative spectrogram of dmPFC LFP during EXT1 (top), displaying a 4 Hz component induced by the CS. 2–6 Hz filtered LFP traces from dmPFC (blue), showing the apparition of 4 Hz oscillations after the onset of the CS (red rectangle). **b** A 4 Hz component (2–6 Hz) is visible in the Power Spectrum Density (PSD) dmPFC LFP during EXT1 for CT + SAL (left) and Gi+CNO (right), average ± SEM, dashed lines represent the 4 Hz range (2–6 Hz). **c** The fraction of the PSD representing 2–6 Hz oscillations is increased during Extinction compared to Baseline. Wilcoxon test, ***$p$ < 0.001. The fraction of the PSD during CS is increased in Gi + CNO compared to CT + SAL, Mann–Whitney $U$ test, ##$p$ < 0.01 (CT + SAL = 14 recording sites from 4 mice, Gi + CNO = 15 recording sites from 5 mice). **d** Average firing rate of a MD neuron during EXT1, centered on the onsets of CSs (top left), binned average amplitude 4 Hz oscillations in the dmPFC (top right). Example raster and corresponding trace of amplitude 4 Hz oscillations in the dmPFC, bursts are displayed in red (bottom left). Scatterplot displaying the averaged firing rate (1 s

bins) as a function of the corresponding average amplitude 4 Hz oscillations, Linear regression line and confidence interval are shown in red (bottom right). Pearson's correlation coefficient, ***$p$ < 0.001. **e** Distributions of Pearson's correlation coefficient of MD neurons firing rate and dmPFC 4 Hz amplitude for CT + SAL (top) and Gi + CNO (bottom). **f** Modulation of burst occurrence of MD neurons firing during episodes of high dmPFC 4 Hz. Mann–Whitney $U$ test, #$p$ < 0.05. Wilcoxon test, ***$p$ < 0.001. **g** Distributions of burst occurrence of MD neurons firing during episodes of high dmPFC 4 Hz during CS. Mann–Whitney $U$ test, ###$p$ < 0.001. (CT + SAL = 39 neurons recorded from 4 mice, Gi + CNO = 67 neurons recorded from 5 mice). Boxplots represent quartiles and whiskers correspond to range; points are singled as outliers if they deviate more than 1.5 x interquartile range from the nearest quartile. Data available at doi:10.5061/dryad.9kd51c5ng. All tests are two-sided. Detailed statistical results are available in the Supplementary Tables referenced by panel numbers.

show that the MD bursting is temporally organized by dmPFC 4 Hz oscillations during fear extinction, and that the inhibition of FN-MD pathway during extinction increases the phase locking of MD bursting to dmPFC 4 Hz oscillations.

While these results suggest that MD bursting is entrained by dmPFC 4 Hz oscillations, we investigated whether the occurrence MD bursting would reciprocally impact on the amplitude of the later dmPFC 4 Hz oscillations. The comparison of 4 Hz oscillation amplitude 500 ms before and after MD bursting during CS presentations revealed that the amplitude of dmPFC 4 Hz oscillations after MD bursting was significantly higher than before bursting if the bursting occurred during episodes of high dmPFC 4 Hz, in both experimental groups (Fig. 7d). Indeed, this increase in amplitude was higher if the bursting occurred during episodes of high dmPFC 4 Hz compared to episodes of low dmPFC 4 Hz, as displayed by an increased ratio of dmPFC 4 Hz amplitude after/before MD bursting (Fig. 7e). In the case of MD bursting occurring during episodes of

low dmPFC 4 Hz, the amplitude after burst was not significantly different from the amplitude preceding the burst (Supplementary Fig. 5b). Thus, these data suggest that MD bursting during episodes of high dmPFC 4 Hz contributes in both groups to the maintenance of these oscillations.

**dmPFC-MD 4 Hz coherence is modulated by the FN inputs to MD**
Since our data showed that dmPFC 4 Hz oscillations can organize MD neuronal activity, we then investigated the presence of an oscillatory 4 Hz component in the MD LFP. Interestingly, similar to our previous observations in the dmPFC, the presentation of the CS during fear extinction resulted, in both experimental groups, in high levels of 4 Hz oscillations in the LFP of the MD synchronized with the dmPFC oscillations (Fig. 8a). This was evidenced by the presence of peaks in the Power Spectrum Density (PSD) in the 4 Hz range (2–6 Hz) (Fig. 8b), and by an increased fraction of the PSD associated to 4 Hz oscillations when compared to baseline (Fig. 8c).

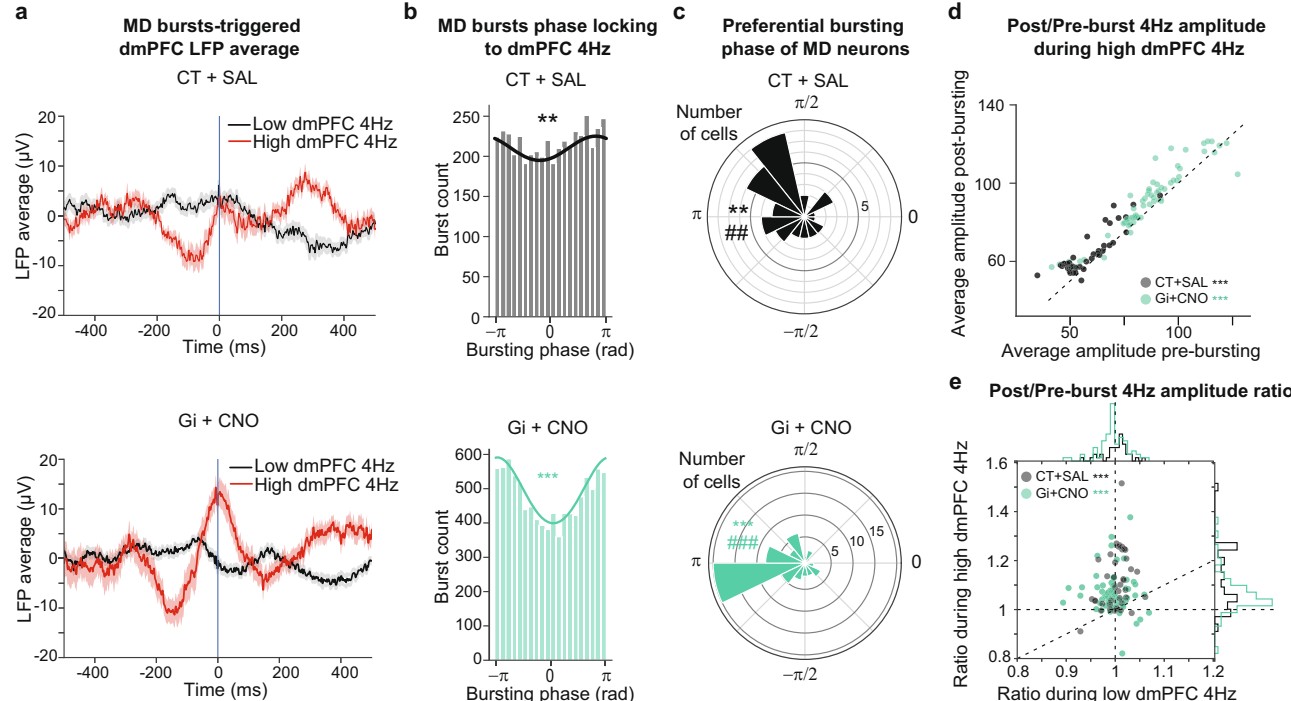

**Fig. 7 | Phase-locking of MD bursting to dmPFC 4 Hz LFP oscillations during EXT1 is increased by chemogenetic inhibition of FN-MD. a** Average of dmPFC LFP (average ± SEM), around MD bursts occurring during episodes of either high dmPFC 4 Hz (red) or low dmPFC 4 Hz) red, for CT + SAL (top) and Gi+CNO (bottom). **b** Phase histograms of MD bursting during episodes of high 4 Hz, for CT + SAL mice (top) and Gi + CNO mice (bottom). The solid line represents a Von Mises fit associated to the circular distribution. Rayleigh test **$p < 0.01$, ***$p < 0.001$. **c** Distributions of preferential bursting phases of MD neurons during episodes of high 4 Hz, for CT + SAL mice (top) and Gi+CNO mice (bottom). Rayleigh test **$p < 0.01$, ***$p < 0.001$. Circular V test, testing for a unimodal circular distribution centered on pi (which corresponds to positive peaks in dmPFC 4 Hz oscillations) ##$p < 0.01$, ###$p < 0.001$. **d** The average amplitude

of dmPFC 4Hz oscillations during the 500 ms following MD bursting is increased compared to the 500 ms preceding the burst during high dmPFC 4 Hz episodes. **e** The ratios of the average 4 Hz amplitude during the 500 ms after and before MD bursting are higher during episodes of high dmPFC 4 Hz compared to episodes of low dmPFC 4 Hz, vertical and horizontal dotted lines are centered on 1, the diagonal dotted line corresponds to equal ratios in both conditions. Distributions of ratios in both conditions are displayed in the marginal histograms. Wilcoxon test, ***$p < 0.001$. (CT + SAL = 39 neurons recorded from 4 mice, Gi + CNO = 67 neurons recorded from 5 mice). Data available at doi:10.5061/dryad.9kd51c5ng. All tests are two-sided. Detailed statistical results are available in the Supplementary Tables referenced by panel numbers.

4 Hz filtered LFP traces (2–6 Hz) from the dmPFC and MD recording sites were strongly correlated in both control and FN-MD inhibited groups (Fig. 8e), with maxima in the cross-correlograms present at short negative lags (Fig. 8e), suggesting that dmPFC 4 Hz oscillations entrain MD 4 Hz oscillations. This observation was confirmed by the presence of a sharp 4 Hz peak in the Generalized Partial Directed Coherence (GPDC) in dmPFC-to-MD direction (Fig. 8f). Furthermore, the 4 Hz GPDC in the dmPFC-to-MD direction was increased in the FN-MD inhibition group compared to the control (Fig. 8g), indicating that the inhibition of the FN-MD pathway increases the contribution of dmPFC to MD 4 Hz oscillations.

Since our data showed that MD bursting was followed by an increase in dmPFC 4 Hz amplitude, we then investigated the effect of MD bursting on the coherence between the MD and dmPFC 4 Hz amplitude. Our study revealed a transient increase in MD-dmPFC 4 Hz coherence after MD bursting during periods of high dmPFC 4 Hz oscillations (Fig. 9a, b). Indeed, when comparing the average baseline coherence preceding the burst (1000–250 ms before the burst) between the two treatments, we observed higher levels of 4 Hz MD-dmPFC coherence in Gi + CNO mice compared to CT + SAL (Fig. 9c), but MD bursting was followed in both cases by a short-term increase in cortico-thalamic coupling for a duration of about one cycle of 4 Hz oscillations (Fig. 9d). Overall, these data further support that MD bursting participates to the maintenance of dmPFC 4 Hz oscillation via MD-dmPFC synchronization.

## Discussion

The cerebellum has been known to influence fear learning and fear expression[3,35], review in ref. [37], and it may participate in a variety of other emotional processes[47] via the projections of the FN to multiple brain regions, including limbic structures such as the vlPAG, thalamus and hypothalamus[3,5,48,49]. Furthermore, recent evidence revealed that the cerebellum participates in fear extinction in humans and rodents[3,10–12]. However, the circuits relaying the cerebellar contributions to this emotional learning have remained elusive.

In the current study, we identified cerebellar mechanisms regulating fear extinction through a cerebello-thalamo-cortical pathway. Our findings reveal the existence of a functional disynaptic pathway from the cerebellum to the dmPFC through the thalamic MD: we showed that the cerebellar FN projects to the lateral subregion of thalamic MD, which is reciprocally connected to the dmPFC, and that trans-synaptic experiments from the FN results in labeling of calbindin-positive MD neurons that project to the dmPFC. In addition, optogenetic stimulations of FN input to MD elicit fast activation of MD, consistent with a direct pathway. Interestingly, while non-specific optogenetic stimulation of the FN triggered fast activation of some dmPFC neurons, we did not find fast responses when the stimulation was confined to the MD-projecting FN neurons. This absence (or scarcity) of fast mPFC response is consistent with previous results showing that MD activation may only increase the firing rate in fast-spiking mPFC neurons, and with a modulatory role of the MD inputs on

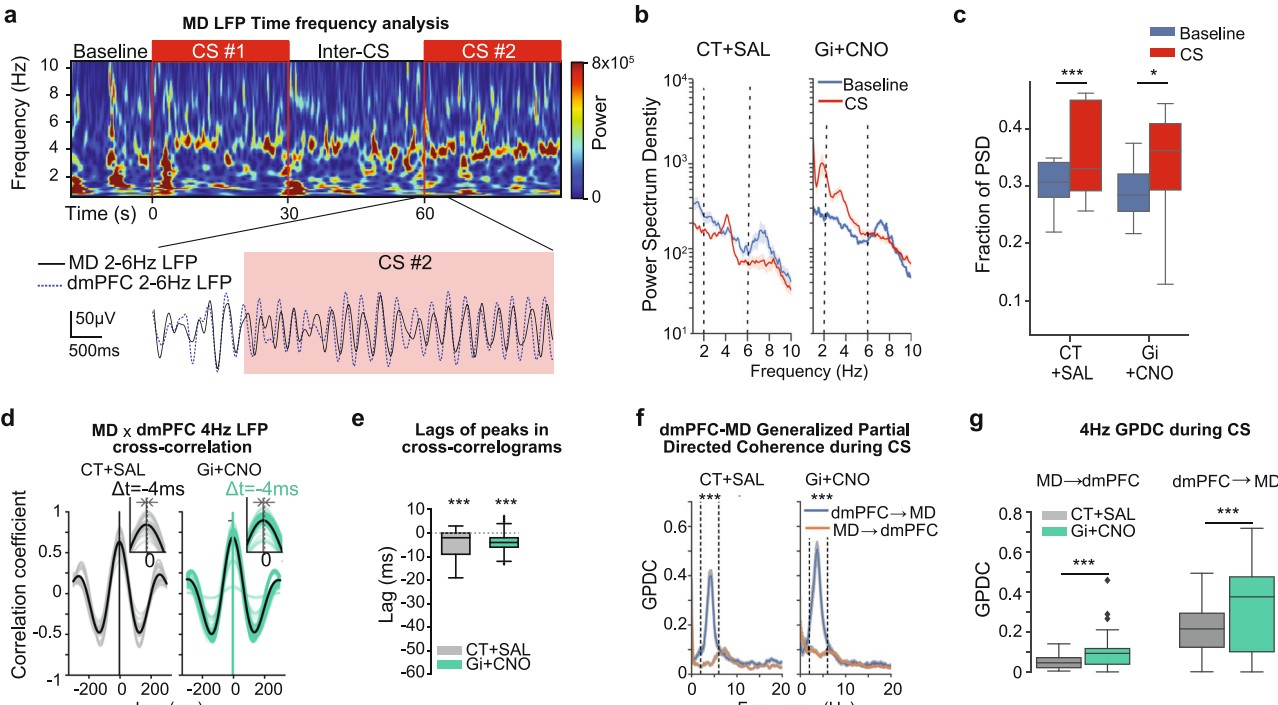

**Fig. 8 | dmPFC-MD 4 Hz coherence is increased by chemogenetic inhibition of FN-MD. a** Representative spectrogram of MD LFP during EXT1 (top), displaying a 4 Hz component induced by the CS. 2–6 Hz filtered LFP traces from dmPFC (blue), showing the apparition of 4 Hz oscillations after the onset of the CS (red rectangle). **b** A 4 Hz component (2–6 Hz) is visible in the Power Spectrum Density (PSD) MD LFP during EXT1 for CT + SAL (left) and Gi + CNO (right), average ± SEM, dashed lines represent the 4 Hz range (2–6 Hz). **c** The fraction of the PSD representing 2–6 Hz oscillations is increased during Extinction compared to Baseline. Wilcoxon test, ***$p < 0.001$ (CT + SAL = 28 recording sites from 4 mice, Gi + CNO = 30 recording sites from 5 mice). **d** Cross-correlation of 2–6 Hz filtered LFP in MD and in dmPFC display maxima at negative lags. Colored lines represent pairwise cross-correlations. Black lines represent averages. **e** Lags of maximum peaks in cross-

correlations, Wilcoxon test ***$p < 0.001$. **f** Generalized Partial Directed Coherence (GPDC) between dmPFC and MD LFP during CS, Average ± SEM, Wilcoxon test on the 4 Hz GPDC dmPFC->MD versus MD->dmPFC, *$p < 0.05$, **$p < 0.01$, ***$p < 0.001$. **g**, 4 Hz GPDC dmPFC -> MD. Mann–Whitney $U$ test, *$p < 0.05$, **$p < 0.01$, ***$p < 0.001$ (CT + SAL = 28 recording sites in MD and 14 recording sites in MD from 4 mice, Gi+ CNO = 30 recording sites in MD and 15 recording sites in the dmPFC from 5 mice). Boxplots represent quartiles and whiskers correspond to range; points are singled as outliers if they deviate more than 1.5 x interquartile range from the nearest quartile. Data available at doi:10.5061/dryad.9kd51c5ng. All tests are two-sided. Detailed statistical results are available in the Supplementary Tables referenced by panel numbers.

mPFC activity in vivo[42]. Another pathway linking the cerebellum and mPFC via the ventral anterior lateral thalamus has been previously hypothesized to explain the cerebellar control of mPFC discharge associated with the conditioned responses in trace eyeblink conditioning[50]. More recently, functional connectivity between the cerebellum and the mPFC has been shown to influence social and repetitive/inflexible behaviors through the ventromedial (VM) thalamus[51]. Both VM and MD are matrix-type thalamus, but their contribution to mPFC activity is functionally distinct since inputs from these two thalamic nuclei target different cell types and layers in the mPFC[52,53].

Cerebellar projections to sub-regions of the MD seem to be conserved across mammalian species, reviewed in ref. [54], and have been proposed to contribute to the cerebello-prefrontal interactions, reviewed in ref. [55]. The FN-MD pathway described in our study is well-positioned to relay cerebellar influence on extinction learning, since cerebellar territories associated to fear have been found in the medial part of the cerebellum[7,8]. In addition, the MD has bidirectional connections with mPFC regions[21,41,56,57], specially the ACC, PL and IL, key brain structures for fear extinction[16–20,27–29], and MD controls dmPFC synaptic plasticity required for fear extinction[25,26].

A remarkable finding in our study is that the specific inhibition of FN input to the MD leads to an impairment of the extinction learning. This effect is primarily effective when starting extinction from high fear expression levels and does not reflect a simple prevention of the expression of fear extinction learning since it failed to affect the fear response when performed only in EXT3. These results suggest that the

FN facilitates fear extinction when fear responses are high. Indeed, we found that inhibition of FN-MD projections affects the firing pattern of MD neurons, and notably increases the bursting activity during the CS. While MD bursting, also observed between CS, does not seem to cause freezing, it has been shown to prevent extinction learning. Moreover, while coordinated activities between limbic brain structures are important for fear extinction[33], our work reveals that MD bursting is regulated by the limbic 4 Hz oscillations that take place during fear expression. The dmPFC plays a central role in these oscillations[30], which are synchronized with respiration via the olfactory bulb[46], and regulated by the amygdala[33,45,58]. Interestingly, we also found that the inhibition of this cerebellar output enhances the entrainment of MD bursting to dmPFC 4 Hz oscillations. This is consistent with the role of the cerebellum in modulating brain oscillations and their coordination across different brain regions[59–61], review in ref. [62], as with a modulation of slow oscillations (<10 Hz) in the mPFC[63,64]. In addition, our data show that the amplitude of the dmPFC 4 Hz oscillations is increased after MD bursts, suggesting that the bursts exert a positive feedback participating in the maintenance of these oscillations.

In agreement with previous trans-synaptic work[5], we found that the FN preferentially targets the dmPFC rather the vmPFC via the MD. While extinction learning has been tightly linked to the vmPFC[16,17], failure to extinct fear responses is linked to enhanced dmPFC activity[65] and the dmPFC 4 Hz oscillations play a central role in the maintenance of the fear response[46,65]. Therefore, by dampening the entrainment of MD bursting by dmPFC oscillations, the cerebellum –under physiological conditions- may limit the positive feedback

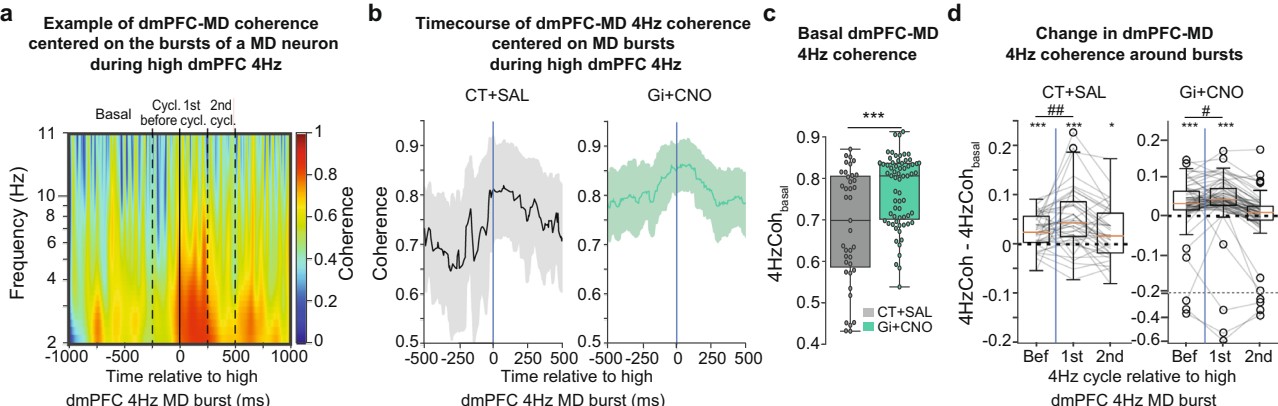

**Fig. 9 | MD bursting is followed by high 4 Hz MD-mPFC coherence. a** Example of time frequency coherence analysis between dmPFC and MD LFP, centered on the bursting of a MD neuron during episodes of high 4 Hz showing an increase of coherence in the 4 Hz range after the burst. **b**, Average 4 Hz coherence between dmPFC and MD LFP, centered on MD bursting during episode of high 4 Hz for CT + SAL mice (left) and Gi + CNO mice (right), showing an increase of coherence in the 4 Hz range after the burst. **c**, Distributions of basal coherence between dmPFC and MD LFP before MD bursting (−1000 ms to −250 ms) during episodes of high 4 Hz. Mann−Whitney *U* test, ***p < 0.001. **d** Distributions of increase in coherence between dmPFC and MD LFP during episodes of high 4 Hz, before MD bursting (−250 ms to 0 ms) and during the first and second 4 Hz cycle after bursting. One sample Wilcoxon test, *p < 0.05, ***p < 0.001. Two samples Wilcoxon test, #p < 0.05, ##p < 0.01 (CT + SAL = 39 neurons recorded from 4 mice, Gi + CNO = 67 neurons recorded from 5 mice). Boxplots represent quartiles and whiskers correspond to range; points are singled as outliers if they deviate more than 1.5 x interquartile range from the nearest quartile. Data available at doi:10.5061/dryad.9kd51c5ng. All tests are two-sided. Detailed statistical results are available in the Supplementary Tables referenced by panel numbers.

exerted by MD on dmPFC 4 Hz oscillations and hence on fear responses[30,46]. Indeed, the drop in freezing response observed in Fig. 4e (top panel) between the end of EXT 2 and beginning of EXT3 when releasing the inhibition of FN-MD projections suggest that this inhibition also promotes the expression of the learned fear response. Our results in high-freezing mice suggest that high MD bursting during high levels of fear expression (>70%) could be particularly effective at preventing the expression of extinction learning. Moreover, recent studies have revealed the importance of the neuronal connections and dynamical synchrony between the dmPFC and vmPFC in extinction learning[66,67]. Our data are therefore consistent with a modulatory role of the dmPFC in fear extinction learning, by which the cerebellum exerts a control over fear extinction.

Finally, our demonstration of the role of cerebellum in facilitating the extinction of the fear response is in concordance with the role of the cerebellum in rapidly adapting the behavior to unexpected situations[3,10,68]. Beyond the classical role of cerebellum in tuning motor behavior, our work provides an anatomo-functional substrate for its role in adapting the emotional behavior strategy. Overall, our work provides support for the role of the cerebellum in psychiatric conditions with deficiencies of fear extinction, such as PTSD, reviewed in ref. [14]. Targeting the lateral cerebellum have been recently shown as a relevant target for the treatment of autism-spectrum disorder[69], while stimulation of the medial cerebellum improved the negative symptoms in schizophrenia[70]. Our work may therefore motivate future research on therapeutic approaches targeting the cerebellum or its projections to the thalamus for the treatment of fear-related and anxiety disorders.

## Methods
### Animals
Adult male C57BL/6 N mice (Charles River, France,MSR Cat# CRL_27,RRID:IMSR_CRL:27), were housed in groups of 4 mice per cage, with free access to food and water, maintained at 21–22 °C, with a 12 h:12 h light/dark cycle with a constant humidity of 40%. Adult male wild-type mice 8–12 weeks of age were used for all the experiments. Male and female mice are equally suitable for fear conditioning analysis[71], but male mice are bigger and have therefore more strength to carry the implant and preamplifier in electrophysiological experiments. Animal care and experimental

procedures followed the European Community Council Directives (authorization number APAFIS#1334-2015070818367911 v3 and APAFIS #29793-202102121752192).

### Stereotactic surgeries
Surgeries for tracing, chemogenetic, optogenetic and electrophysiological experiments, were performed in 7–8 weeks-old mice, placed in a stereotaxic frame (Kopf Instruments) after buprenorphine administration, and anaesthetized with isoflurane 2% along the whole procedure. In addition, a local analgesia was administered subcutaneously above the skull with 0.02% of lidocaine. The body temperature was monitored and maintained at 36 °C with a heating pad and a rectal thermometer.

For viral injections and electrodes and optic fiber implantations, the stereotaxic coordinates used for FN (AP: −6.37, ML: ±0.70, DV: −2.12), MD (AP: −1.12, ML: ±0.65, DV: −3.15) and dmPFC (AP: +2.34, ML: ±0.25, DV: −1.5) were taken relative to the bregma, and the depth was considered from the brain surface. Glass capillaries used for viral infusions were kept at the injection site for 5 min after viral infusion before being slowly withdrawn.

### Neuroanatomical tracing and histology
For neuroanatomical tracing a group of mice was injected with 100 nl of anterograde AAV1-CB7-Cl-mCherry-WPRE-rBG (Upenn Vector) in the FN (n = 6), 100 nl CTB-alexa 488 (Invitrogen) in the dmPFC (n = 6), or retrograde AAV-Syn-eGFP (Addgene) were injected in the MD (n = 6). For trans-synaptic labeling, a group of mice was injected with 75 nl of AAV9-CMV-PI-Cre-rBG (Addgene) in the FN, and 100 nl of AAV1-CAG-Flex-tdTomato-WPRE-bGH in the MD (n = 3), and MD sections were used to perform Calbindin immunostaining.

Mice were deeply anesthetized with ketamine 80 mg kg⁻¹ and xylazine 10 mg kg⁻¹ i.p., and transcardially perfused with formalin (Sigma) 3–4 weeks after the surgery to assess the viral expression. Brains were dissected and kept overnight in formalin at 4 °C and then placed in PBS solution until slicing. Coronal brain sections of 50 or 90 μm were made using a vibratome (Leica VT 1000 S), then dried and mounted with Mowiol (Sigma) or Fluoroshield with DAPI medium (Sigma). The slices were analyzed and imaged using a confocal microscope (Leica TCS Sp8).

The same procedure was used to verify electrode and optic fiber implantation sites and viral expressions for each experiment, mice were perfused as detailed above, and mice with no viral expression or misplacement of the viral infusion, optic fiber or electrode were excluded from the analysis. Electrode placements are summarized in the Supplementary Fig. 6.

## Immunohistochemistry

For the immunostaining, 50 μm slices containing MD were washed with PBS-0.3% Triton X-100, blockage of nonspecific sites was assessed by 2 h of incubation with 3% normal donkey serum (NDS). Sections were then incubated in a solution containing monoclonal mouse anti-Calbindin-D-28K (1:300, Sigma, C9848) in PBS-0.3% Triton X-100 with 1.5% NDS at 4 °C for 24 h. After first antibody incubation, slices were rinsed and sections were incubated at room temperature for 2 h with secondary antibody, donkey anti-mouse IgG conjugated to Alexa Fluor 488 (1:300, Invitrogen, R37114). Slices were mounted with Mowiol and analyzed with the confocal microscope.

## Fear conditioning and extinction protocol

Previous to behavioral experiments, mice were daily handled during one week. Fear conditioning and extinction protocol were performed in two different chamber configurations (Context A or B), placed inside of a sound-attenuating box (Ugo Basile). The fear conditioning was performed in the Context A, which consisted in a rectangular-shaped Plexiglas chamber, $17 \times 17 \times 25$ cm (L × W × H) dimension, with a grid floor and black and white checker walls, scented with peppermint soup. On fear conditioning day (day 1), mice were left for 5 min to habituate to context A, then they were exposed to 5 presentations of a tone (30 s, 80 dB, 2.7 kHz, representing the CS), that co-terminated with a mild foot-shock (0.5 sec, 0,4 mA, representing the US), and with an interval between CS-US presentations of 120 s. During the following 3 days, mice underwent the fear extinction training, which consisted in 3 days of extinction sessions (EXT1-3). Extinction sessions were performed in the Context B, composed by a yellow cylindrical chamber (17 cm diameter, 25 cm H), grey Plexiglas floor, and vanilla scented. Each extinction session started with 5 min of habituation period in the novel context B, followed by 25 consecutive un-reinforced 30s-long CS presentations, with an interval between CS of 30 s.

Mice were video-tracked using a high-definition video camera, positioned above the testing chambers. Stimuli administration was controlled by the EthoVision XT 14 software (Noldus Information Technology), which assessed also the freezing behavior during the experiments as inactive periods (the threshold of inactivity was set to not detect the breathing movements as active periods). All mice returned to their home cage after the experiments. Analysis of freezing levels in each session was performed by comparisons between groups of the first CS and CS5 for FC, and averaged of CS1-CS5 representing "early" EXT, CS10-CS14 as "middle" EXT, and CS21-CS25 as "late" EXT.

## Anxiety tests

Open field, elevated plus maze and dark-light box tests were performed in order to analyze anxiety-like behavior. On week before subjecting the mice to fear conditioning, they were subjected on successive days to anxiety tests, starting with the less anxiogenic, open field test, followed by elevated plus maze and dark-light box. Open field test consisted in a 38 cm diameter circular arena (Noldus), under 40–50 lux luminosity. Each mouse was placed in the center of the arena and allowed to explore freely for 10 min. Frequency of entries to the center of the arena, time spent in center, distance moved in center, were measured. Elevated plus maze test was realized using an elevated (52 cm above the floor), plus-shaped apparatus (Noldus) with 2 open arms (36 ×6 cm L × W) and 2 closed arms (36 × 6 × 25 cm L × W × H), under 40–50 lux luminosity in the open arms. Mice were left to freely explore the arena for 5 min, while assessing the frequency of entries in

open and closed arms, time spent in open arms, total distance moved. In the dark-light box test, each mouse was placed in the light zone (luminosity 500–600 lux, $40 \times 20 \times 20$ cm L × W × H dimension) of the apparatus (Noldus) and left to freely explore for 5 min, while assessing the frequency of entries in light zone, time spent in light zone, latency to enter in dark zone (0 lux, $20 \times 20 \times 20$ cm L × W × H dimension), and total distance moved. Mice were video tracked and behavior was analyzed using EthoVision XT 14.

## Hot plate and tail immersion tests

Pain sensitivity was analyzed using hot plate and tail immersion assays performed two weeks after the end of the other experiments. Mice were placed on a hot plate (55 °C) (Harvard apparatus), until they jumped or licked their hind paw. The latency to the first reaction was measured. After the hot plate test, tail immersion test was performed. The mice tail tips were immersed in hot water (50 °C) and the latency of the tail flick or withdrawal was measured. All experiments were video tracked.

## Chemogenetics

Specific chemogenetic inhibition of the FN neurons projecting to MD was carried out by bilateral injection of 250 nl of a retrograde virus expressing a cre-recombinase in the MD (CAV2-cre-GFP, from Plat-forme de Vectorologie de Montpellier) in combination with the infusion of 200 nl of inhibitory cre-dependent DREADD (AAV-hSyn-DIO-hM4Di-mCherry, Addgene) bilaterally in FN, adapted from previous works in our team[72] which demonstrated a strong reduction in cerebellar nuclei firing rate upon injection of CNO (1 mg/kg). The behavioral experiments started 10–14 days after the surgery, to ensure the expression of the receptors and the recovery of the mice. Transient inhibition of the FN MD-projecting neurons was assessed by intraperitoneal administration of the clozapine N-oxide (CNO, Tocris Bioscience) dissolved in saline solution (1.25 mg/kg), while a control group received only saline administration, 30 min before the first 2 extinction sessions (EXT1 and EXT2) or 30 min before anxiety or nociceptive tests.

Additional controls were performed to verify the specificity of the pathway inhibition and the DREADD-modulation effect. To corroborate that the CNO dose has not an effect per se in our experimental conditions, a batch of mice (named "Sham" mice) underwent the same surgical procedure with the injection of AAV-hSyn-mCherry instead of the DREADD vector. During behavioral experiments, the Sham mice received either an intraperitoneal injection of saline or CNO solution (1.25 mg/kg), 30 min before the tests, and freezing levels were analyzed in absence of DREADD expression. On the other hand, to verify that the effect observed under DREADD inhibition was specific of the FN terminals in MD, mice were implanted with 30 G cannulas bilaterally in the MD to allow the local infusion of 200 nl of CNO or saline intracranially. During extinction sessions, mice expressing inhibitory DREADD received either sterilized filtered PBS or CNO (0.5 mM) infusion, and another group of Sham mice received CNO (0.5 mM). The intracranial infusions were performed 10 min before the beginning of each session, by using a pump at an infusion speed of 100 nl/min, and a total volume of 250 nl. Freezing levels were analyzed and the positions of the cannulas were corroborated histologically.

## Electrophysiological recordings

Extracellular recordings were assessed using standard 16 channels electrode interface boards (EIB-16; Neuralynx), to which 2 cannulas were attached, one for each recorded region, MD and dmPFC. The electrode bundles consisted of nickel chrome wires (16μm diameter, Coating ¼ Hard PAC, KANTHAL Precision Technology) twisted in groups of 6, gold-plated to 100–400 kΩ (cyanure-free gold solution, Sifco). Two bundles were inserted per cannula (stainless steel, 30 Gauge, Phymep). The recording electrodes were then progressively

lowered until they reached the targeted brain structures and the electrode interface boards were cemented to the skull (Dental Parkell Adhesive Resin Cement Super-Bond C&B). Miniature stainless steel screws were implanted on the left parieto-occipital suture, serving as electrical reference and ground. The skin ridges were sutured and mice were allowed to recover in their home cage for at least 10 days. Recordings were performed on a TDT system control by Synapse v95 (Tucker-Davis Technologies, Davis, CA) when no fear conditioning was involved, or on Multi-Channel System W2100 system with their recording software (v1.5.6)

### Optogenetics and electrophysiological recordings

The neural activity in the MD and dmPFC was recorded during the optogenetic activation of the contralateral FN neurons. Expression of channelrhodopsin 2 (ChR2) was assessed by injection of an anterograde AAV8-Syn-ChR2-H134R-EYFP (Addgene) in the FN, and an optical was fiber implanted above (200 μm diameter, 0.22 aperture, fixed in a stainless steel ferula, Thor Labs) and contralateral to the recording sites. Electrodes were placed through cannulas in the MD and dmPFC and the EIB were fixed to the skull. The optical stimulations were performed three or four weeks after the surgery (to ensure the ChR2 expression), in freely moving mice. After a 5 min habituation period in an open field (38 cm diameter), the activation of the FN projections was done, using light pulses of 100 ms, at 0.25 Hz, 1 mW, for 30 min (with a 2 min break every 10 min block). The choice of light intensity to stimulate FN neurons was based on the calibration performed in our previous work[3]. The activation of MD and dmPFC neurons by optogenetic stimulation of FN neurons was assessed by computing the Peri-Stimulus Time Histogram (5 ms bins) around the optogenetic stimulation. This PSTH was then normalized using a classical Z-score (subtracting by the mean value of the PSTH during the 100 ms before the stimulation and dividing it by its standard deviation). Normalized PSTH reaching an absolute Z-score value superior to 3 during the 100 ms stimulation were considered responsive.

In another group of mice, the specific activation of the FN-MD pathway was assessed by injecting a cre-dependent anterograde adenovirus expressing ChR2 (AAV-Dio-ChR2-EYFP, Addgene) in the FN, together with a retrograde adenovirus expressing cre-recombinase (AAV-Cre-mCherry, Addgene) in the contralateral MD (left side). The implantation and surgical procedures were identical with the ones described above. The FN MD-projecting neurons were stimulated by pulses of 10 ms, at 0.5 Hz and 1 mW, during 10 min. Luminous stimuli were administered and neural signals were recorded using Tucker Davis Technology System 3 acquisition system (25 Hz sampling rate, RZ2, RV2, Tucker-Davis Technologies) and the spike sorting was performed using Matlab scripts. Viral infusions and implant positions were confirmed histologically when the experiment ended. Analysis of the spike sorted data was performed using Python. The specific activation of MD and dmPFC neurons by optogenetic stimulation of FN MD-projecting neurons was assessed by computing the Peri-Stimulus Time Histogram (5 ms bins) around the optogenetic stimulation. This PSTH was then normalized using a classical Z-score (subtracting by the mean value of the PSTH during the 100 ms before the stimulation and dividing it by its standard deviation). Normalized PSTH reaching an absolute Z-score value superior to 3.5 were considered responsive.

In order to assess the validity of the optostimulation experiments, we performed a control experiment to verify that FN illumination in absence of ChR2 expression does not affect spike firing in the areas recorded. We implanted an optic fiber in the FN together with electrodes in FN, MD and dmPFC, in mice injected with retrograde AAV-GFP (without the expression of ChR2) in the MD, and performed 100 ms illuminations at 0.25 Hz, 1 mW, for 30 min (with a 2 min break every 10 min block), while recording in the FN, MD and dmPFC (Supplementary Fig. 2A–B). There was no variation of firing rate at the population level in the FN, MD and dmPFC (Supplementary Fig. 2C),

although we observed 1 neuron in the FN and 3 neurons in the MD reaching the threshold of significant variation of firing rate during the 100 ms of illumination (Supplementary Fig. 2D). However, the transient nature of the mild increase in firing rate suggests a coincidental classification as responsive cells.

Since the wireless transmitter used in the electrophysiological experiments represented a significant hindrance for the movement and was associated with atypical immobile postures (nose down and/or tilted head to rest the preamplifier on the floor or against the wall) which could indifferently reflect freezing or resting, we could not score freezing in these mice.

### Chemogenetics and electrophysiological recordings

Mice expressing cre-dependent inhibitory DREADDs in FN MD-projecting neurons (see chemogenetics section) were implanted with electrodes in the MD and dmPFC (as described in "Electrophysiological recordings" section). The neuronal activity of the MD and dmPFC was examined during FC and extinction learning, under chemogenetic inhibition of the FN-MD projections during EXT1 and without manipulation during EXT3. After experiments were concluded, viral expression and electrode positions were confirmed by histology.

### Quantification and statistical analysis

**Behavioral data analysis.** For freezing analysis, the data were analyzed by repeated-measure ANOVA computed with R 3.6.3 (lme package version 3.1-3) and posthoc tests were performed with the package emmeans (version 1.7.0) using Rstudio 4.1.2. During extinction, Early, Middle, Late values of freezing respectively correspond to the average of the 5 first, 5 middle and 5 last CS freezing scores. For other behavioral measures, $t$-test were used (Graph Pad Prism® version 7). All tests used are two-sided (when applicable).

### Electrophysiological data analysis

**Burst analysis.** We quantified the occurrence of bursts in MD, detected using the Robust Gaussian surprise algorithm 1. Spike trains were processed in the following way: Inter-Spike Intervals (ISI) were calculated, then transformed to log (ISI). A central set of ISI was defined as the portion of this distribution lying under [E−1.64*MAD; E + 1.64*MAD], where E is the midpoint of top and bottom 100*p percentile of the log (ISI)s with $p = 0.05$. The central location of this distribution was defined as the median of the central set previously mentioned, and the distribution of log (ISI) was then normalized by subtracting the central location. This normalization process was performed on the entire spike train using a sliding window of half-width 0.2*N/2, where N is the number of spikes in the spike train. The burst-threshold is set as 0.5 percentile of the central distribution. Bursts seeds were defined as normalized ISI being below the burst-threshold. Those seeds were then extended by recursively trying to add the previous and the next normalized ISI, if the addition lead to an increase of the associated $p$-value, the concatenation process was stopped, otherwise it was continued. Overlapping bursts strings were cleaned by keeping the strings with the lowest associated $p$-value, thus making them mutually exclusive. Burst occurrence was computed as the number of bursts strings divided by the duration of the recorded spike trains. The average burst firing rate was calculated as the number of spikes contained within a burst divided by the total duration of burst strings in the spike train.

**Local field potential, spectral analysis and phase locking.** In order to process the LFP, the raw electrophysiological traces sampled at 25 KHz were filtered between 1 Hz and 200 Hz, then downsampled to 1 kHz. All signals were filtered using the filtfilt function from the package scipy, with zero-phase distortion 5th order Butterworth filters. Power Spectral Densities (PSD) were computed using the welch function from the scipy package. The fraction of the PSD representing 4 Hz

oscillations was defined the integral of the PSD in the 4 Hz range (2–6 Hz) divided by the integral of the PSD from 1 to 100 Hz. To assess the relationship between LFP in the dmPFC and in the MD, Generalized Partial Directed Coherence analysis was performed on the processed LFP traces using the package spectral_connectivity from Eden Kramer Lab (https://github.com/Eden-Kramer-Lab/spectral_connectivity). For the specific analysis of 4 Hz LFP, the processed LFP traces were filtered in the desired frequency band (2–6 Hz) using the filtfilt function from the scipy package, with zero-phase distortion 5th order Butterworth filters. To study the relationship between the timing of MD neuronal activity and the dmPFC 4 Hz oscillations, Hilbert's transform was performed using the function hilbert from the package scipy, allowing the extraction of the instantaneous phase and amplitude of envelope estimated at every sample point of the signal. Phases were expressed in radians, with a phase of 0 corresponding to a negative peak in the 4 Hz LFP. Phase locking analysis of bursting activity was performed by considering the first spike of each bursts. Considering spikes occurring in periods associated to weak 4 Hz oscillations would result in the inclusion of uninformative phases in the analysis. So, in order to prevent this bias, only burst occurring in periods of high 4 Hz were considered. Periods of high 4 Hz were defined as periods were the amplitude of envelope was superior to the median amplitude of envelope during the baseline before the Extinction session plus one median absolute deviation (MAD). In other words, we considered the periods where the Robust Z-score of the amplitude of envelope was superior to 1, creating a spike train corresponding to MD bursting during periods of high 4 Hz oscillations in dmPFC. Von Mises distributions fit on phase distributions were performed using the function vonmises from the package scipy, allowing the computation of the parameter Kappa, indicative of the concentration of a circular distribution. Rayleigh tests and circular V-tests with a preferred direction of pi were performed using the package pingouin, respectively using the functions circ_rayleigh and circ_vtest. In order to assess the relationship between MD bursting and the coherence between dmPFC and MD LFP, we used the function spectral_connectivity included in the package mne, For this, LFP snippets of 2000ms centered on the bursts of each individual MD neurons were extracted from the channels displaying the highest fraction of the PSD corresponding to 4 Hz oscillations in the MD and in the dmPFC, and the coherence between MD and dmPFC LFP was calculated using Morlet wavelets of the first order, yielding a time frequency coherence analysis centered on MD bursts.

### Reporting summary
Further information on research design is available in the Nature Portfolio Reporting Summary linked to this article.

### Data availability
The data generated during this study have been deposited on the Dryad database with the reference doi:10.5061/dryad.9kd51c5ng.

### Code availability
The source code generated during this study have been deposited on the Dryad database with the reference https://doi.org/10.5281/zenodo.7603770.

### Material availability
This study did not generate new unique reagents.

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

## Acknowledgements

This work was supported by Fondation pour la Recherche Medicale (FRM DPP20151033983 to D.P. and FRM FRM-EQU202103012770 to C.L.) and by Agence Nationale de Recherche to D.P. (ANR-21-CE37-0025 CerebellEmo) and to C.L. (ANR-17-CE37-0009 Mopla, ANR-17-CE16-0019 Synpredict, ANR-21-CE16-0017 PomPom) and by the Labex Memolife and the Institut National de la Santé et de la Recherche Médicale (France). This work was supported by University of Medicine and Pharmacy "Carol Davila", a project number 819/11.01.2019 corresponding to the salary to I.A.G. We thank the Imaging Facility at IBENS (IMACHEM-IBiSA, France-BioImaging ANR-10-INBS-04, FRC Rotary International France, Investments for the future, ANR-10-LABX-54 MEMOLIFE). The authors are grateful to Dagmar Timmann and Thomas Watson for critical reading of the manuscript.

## Author contributions

C.L. and D.P. acquired the funding, designed and supervised the project; J.L.F. performed and analyzed the anatomical and chemogenetic behavioral experiments; R.W.S., I.A.G., H.B.A., and M.A. performed the electrophysiological experiments and data curation (spike sorting, histological verification). R.W.S. and C.L. analyzed the electrophysiological experiments. All authors interpreted results, participated to the writing and approved the final manuscript.

## Competing interests

The authors declare no competing interests.
