## [Peer Review File · Nature Communications]

The cerebellum regulates fear extinction through thalamo-prefrontal cortex interactionsREVIEWER COMMENTS

Reviewer #1 (Remarks to the Author):

This study investigates a role for the fastigial nucleus (FN) to mediodorsal thalamus (MD) to dorsomedial prefrontal cortex (dmPFC) circuit in modulating fear, anxiety, and nociceptive behaviors. The authors identify a novel multi-synaptic circuit from the FN to the MD whose activity is regulated by dmPFC oscillations. Chemogenetic manipulation of FN projections to the MD increased freezing behavior during the extinction of conditioned fear, without affecting on pain- or anxiety-related behavior. The effect of chemogenetic inhibition of FN neurons projecting to MD resulted in greater levels of freezing when the animals were exposed to the CS after CNO administration (Fig 4), but the drug manipulation did not lead to poor extinction retrieval during a drug-free test on the third EXT session. Inhibition of FN->MD projection neurons may have simply increased freezing to the CS, however inhibiting this projection during extinction retrieval had no effect on freezing (supp Fig 3c). Importantly, controls for non-specific effects of CNO indicate that the drug mediates its effects through DREADDs. In addition, the authors found that chemogenetic inhibition of FN projections to MD increased spike bursts in the MD that have previously associated with extinction deficits. The authors further found that 4 Hz oscillations in the dmPFC were time-locked to spike bursts in MD. This study will add great insight into role for cerebellar modulation of forebrain circuits involved in fear conditioning and extinction in the pathogenesis of psychiatric disorders. This paper will be of broad interest to the journal's readership. Here are some points the authors may wish to consider in revision:

- 1) The authors should show baseline (pre-CS freezing) for all of the test sessions insofar as pre-CS freezing will influence assessment of freezing to the CS. It is curious that freezing does not change over the course of EXT1 and EXT2 when systemic CNO is administered to inhibit FN -> MD terminals. Moreover, when the animals are tested off drug, the CNO-treated animals appear to have extinguished to an equivalent degree freezing by EXT3 (see #1). This suggests that the CNO effects are limited to within-session extinction rate (though this is a modest effect), and do not have effects on long-term extinction memory/retrieval. This distinction should be made clear in the manuscript, and the conclusions around the impact of FN input to the MD should be moderated given that it is a relatively time-limited effect.
- 2) Statistical reporting for Fig 4e (EXT2) indicates significant differences (per t-tests) for a drug effect in middle and late blocks, however the omnibus ANOVA (Supp Table 2) does not reveal a significant main effect of drug or drug X block interaction. It is not appropriate to perform post-hoc comparisons on means in an analysis that does not reveal a drug main effect or interaction. Moreover, the absence of a drug effect on EXT2 suggest that the effect is quite modest (EXT1).
- 3) Histological verification of placement of the recording electrodes was not shown. This would be useful for confirming the localization of the probes in the targeted brain areas.
- 4) According to the SAGER guidelines Heidari et al., 2016, Table 1, "If only one sex is included in the study, or if the results of the study are to be applied to only one sex or gender, the title and the abstract should specify the sex of animals or any cells, tissues and other material derived from these and the sex and gender of human participants.". Authors should indicate in the title and abstract that only males were used in the study.

Minor comments.

Line 80 – Could authors provide a magnification for Supplementary Figure 1C.

Line 81 – It isn't clear what the different colors in the atlas picture represents in Figure 1D. Could the authors describe further what these colors mean? Additionally, authors should specify whether the reader is looking at a sagittal or coronal representation of the FN.

Line 183 – Figure legend for panel C is mislabeled.

Line 222 – It isn't entirely clear what the # symbols represent in Figure 5c and Figure 5d. The figure legend indicates that the # symbol is a difference between phases. The authors should specify which phases were different from which.

Line 222 – Additionally, could the authors provide the behavior from the electrophysiological dataset. Prior experiments to this one in the study, CNO was given during EXT1 and EXT2 whereas in this

experiment it was only given during EXT1. It would be interesting to further speculate on the relationship between spike burst occurrences during FN-MD inhibition and behavior.
Line 561- Reasons for performing the experience only in male mice should be specified.
Line 620 – Authors should explicitly specify how long tones were presented during extinction.
Line 631 – It would help readers interpret anxiety-like behaviors if there was a timeline in which these anxiety-like behavioral tests were run. Were they done on consecutive days, all in one day?

Reviewer #2 (Remarks to the Author):

The contribution of the cerebellum to fear conditioning has long been ignored, in particular its contribution to extinction. Only recently, studies have started to investigate the contribution of the cerebellum to extinction of learned fear. This is the very first study investigating how the cerebellum may interact with the very well known cerebral fear extinction network. The study is carefully performed, shows the anatomical connections between fastigial nuclei, thalamus (MD) and prefrontal cortex (dmPFC); and shows that inhibition of this cerebellar-cerebral efferent pathway impedes extinction. Different to what one may expect at first sight (the prelimbic cortex is generally known to be involved in the acquisition whereas the infralimbic cortex is known to be involved in extinction of learned fear responses), the authors show that inhibition of cerebellar output via the thalamus to prelimbic areas impedes extinction. Therefore, findings show that the cerebellum may contribute to extinction primarily via modulation of prelimbic areas. Authors also show how this may happen by demonstrating corresponding changes of thalamic and prefrontal cortex oscillations. The study is innovative (first of its kind) and of major importance for the field.

As stated in the acknowledgments of the paper, the reviewer has seen and commented upon a previous version of the manuscript. I have therefore few comments.

- i) It reads in the manuscript as if anterior cingulate cortex and prelimbic cortex are different areas of the dmPFC. Is this correct, or does ACC refer to the human brain and prelimbic cortex to the rodent brain?
- ii) Out of curiosity: Did the authors investigate whether this cerebellar-thalamic-prefrontal cortex pathway impacts fear acquisition learning, as one would expect?

Dagmar Timmann

Reviewer #3 (Remarks to the Author):

Review of "The cerebellum regulates fear extinction through thalamo-prefrontal cortex interactions" by Frontera et al. In this report, Frontera et al. perform experiments designed to map and perturb a specific Fastigial nucleus to mediodorsal thalamus projection circuit during fear extinction and also while making in vivo recordings from dorsomedial prefrontal cortex (dmPFC) and mediodorsal (MD) thalamus, two areas with reciprocal neuronal connections. They use rigorous mapping experiments with anterograde and retrograde approaches and find very specific circuit connections between caudal FN neurons to specific regions of the MD (contralateral lateral MD) enriched in neurons that are positive for calbindin expression. Next, they perform circuit perturbations with excitatory optogenetics, which induces responses in the MD and dmPFC, consistent with both direct and indirect pathways. These are again quite rigorous, with anterograde and retrograde viral approaches to elicit channelrhodopsin expression, though the authors report that there are differences in results with these two approaches (Figure 3). These experiments show that the FN circuit they are interested in is sufficient to elicit changes in firing in the MD and dmPFC. The authors then use inhibitory chemogenetics to test the role of these neurons in fear extinction and show that inhibition of this

circuit increases freezing in their fear extinction paradigm (blocking the learning of new memory). They show that this manipulation does not alter baseline anxiety, or pain responses. Next, the authors show that chemogenetic inhibition in the FN->MD circuit increases burst firing in the MD. In a real strength of this paper, over the last 3 figures, the authors look at how the FN-MD circuit modulates reciprocal interactions between MD and dmPFC. They confirm that dmPFC shows 4Hz oscillations associated with fear extinction, and that this is correlated with a reduction in MD burst firing, which is reduced when the FN-MD circuit is chemogenetically inhibited in Figure 6. In fig 7, they show that phase locking of the MD bursting and dmPFC 4 Hz LFP oscillation is maintained when the FN-MD circuit is inhibited. In figure 8, they look at the coherence of 4Hz LFP in both MD and dmPFC and show that this is increased by chemogenetic inhibition of the FN-MD circuit.

Strengths:

This is an important, novel and rigorous contribution to the literature, demonstrating a role for a specific FN->MD circuit that is 1) necessary for fear extinction and 2) demonstrates mechanistically, that this circuit modulates reciprocal interactions between MD and dmPFC at the level of both single units and LFPs. This is a major and important contribution to the fear/response to threat literature, as well as the rapidly developing field in cerebellar-mediated cognitive/emotional functions. The data strongly supports the conclusions of the investigators, and I do not detect any major weaknesses in the work. The methodology is sound and is clearly presented.

Minor issues:

The authors have been very careful in controlling for different behaviors that may confound the interpretation of the results. The one control I don't see is whether the inhibition of the FN-MD circuit alters sensorimotor gating at the level of acoustic startle or prepulse inhibition of the startle reflex—if there is a sensorimotor gating deficit, would this alter the interpretation of results?

I'm curious if authors think that the role that the FN->MD circuit fits into a model of how the cerebellum affects behavior? The data strikes me as similar as what one might see in slower cerebellar adaptations to say, the vestibulo-ocular reflex or prism adaptation, as opposed to the rapid corrections it can elicit in other movements. A corollary to this question is whether this circuit plays a role in associative fear learning.

The authors tell us that in Figure 4f, baseline freezing is not different between groups, though I'm very curious about Context A...it looks like there might be an effect of an outlier, which raises concern that the level of baseline freezing is possibly different between groups.

What is the interpretation of why the MD bursting is higher at baseline in the Gi+CNO mice in Fig 5? How were the optogenetic stimulation parameters selected?

Reviewer #1 (Remarks to the Author):

This study investigates a role for the fastigial nucleus (FN) to mediodorsal thalamus (MD) to dorsomedial prefrontal cortex (dmPFC) circuit in modulating fear, anxiety, and nociceptive behaviors. The authors identify a novel multi-synaptic circuit from the FN to the MD whose activity is regulated by dmPFC oscillations. Chemogenetic manipulation of FN projections to the MD increased freezing behavior during the extinction of conditioned fear, without affecting on pain- or anxiety-related behavior. The effect of chemogenetic inhibition of FN neurons projecting to MD resulted in greater levels of freezing when the animals were exposed to the CS after CNO administration (Fig 4), but the drug manipulation did not lead to poor extinction retrieval during a drug-free test on the third EXT session. Inhibition of FN->MD projection neurons may have simply increased freezing to the CS, however inhibiting this projection during extinction retrieval had no effect on freezing (supp Fig 3c). Importantly, controls for non-specific effects of CNO indicate that the drug mediates its effects through DREADDs. In addition, the authors found that chemogenetic inhibition of FN projections to MD increased spike bursts in the MD that have previously associated with extinction deficits. The authors further found that 4 Hz oscillations in the dmPFC were time-locked to spike bursts in MD. This study will add great insight into role for cerebellar modulation of forebrain circuits involved in fear conditioning and extinction in the pathogenesis of psychiatric disorders. This paper will be of broad interest to the journal's readership. Here are some points the authors may wish to consider in revision:

We thank the reviewer for these encouraging comments.

1) The authors should show baseline (pre-CS freezing) for all of the test sessions insofar as pre-CS freezing will influence assessment of freezing to the CS. It is curious that freezing does not change over the course of EXT1 and EXT2 when systemic CNO is administered to inhibit FN -> MD terminals. Moreover, when the animals are tested off drug, the CNO-treated animals appear to have extinguished to an equivalent degree freezing by EXT3 (see #1). This suggests that the CNO effects are limited to within-session extinction rate (though this is a modest effect), and do not have effects on long-term extinction memory/retrieval. This distinction should be made clear in the manuscript, and the conclusions around the impact of FN input to the MD should be moderated given that it is a relatively time-limited effect.

We have included the freezing baseline for all the test sessions as the reviewer requested in Supp Fig 3e-h. We have revised the analysis of the data by showing that the impact of FN->MD inhibition is indeed strong when the initial freezing levels are high (above median), while it is minimal for lower levels of freezing (under the median) ; this is now detailed in Fig 4e. Indeed, the effect of CNO was smoothed out by averaging the freezing of all animals and suggested a modest effect of FN->MD inhibition. However, when only considering the mice which exhibited the strongest fear response (n = 6, typically > 70% of time of freezing in the early stage of EXT1), very little extinction is observed in the course of EXT1 and EXT2 under CNO for this set of mice, while extinction looked similar between groups for the mice with lower fear responses (n=6). Yet, as pointed out by the reviewer some extinction may still take place in later stages (the initial freezing EXT3 was lower than the end of EXT2 for the Gi+CNO group). The possibility of a role of FN-MD projections in preventing

the expression of the learned fear response is now better acknowledged in the Discussion.

2) Statistical reporting for Fig 4e (EXT2) indicates significant differences (per t-tests) for a drug effect in middle and late blocks, however the omnibus ANOVA (Supp Table 2) does not reveal a significant main effect of drug or drug X block interaction. It is not appropriate to perform post-hoc comparisons on means in an analysis that does not reveal a drug main effect or interaction. Moreover, the absence of a drug effect on EXT2 suggest that the effect is quite modest (EXT1).

We thank the reviewer for pointing out this mistake. This prompted the new analysis which revealed that the effect of FN->MD inhibition primarily affects the extinction from high levels of fear memory (point 1 above); the absence of this analysis in the previous version of the MS actually resulted in an apparent lack of effect in EXT2.

3) Histological verification of placement of the recording electrodes was not shown. This would be useful for confirming the localization of the probes in the targeted brain areas.

A summary figure has been added as Supplementary Figure 6. The reassessment of the positioning of electrodes actually led to a small correction in cell numbers (this did not impact qualitatively any of the results). Figure 3 and statistics (lines 112-113) have been updated accordingly.

4) According to the SAGER guidelines Heidari et al., 2016, Table 1, “If only one sex is included in the study, or if the results of the study are to be applied to only one sex or gender, the title and the abstract should specify the sex of animals or any cells, tissues and other material derived from these and the sex and gender of human participants.”. Authors should indicate in the title and abstract that only males were used in the study.

We have indicated the sex of the animals used in this study in the abstract, following the Nature Communication guidelines (“1. If the research findings apply to only one sex or gender, that must be indicated in the title and/or abstract.”)

Minor comments.

Line 80 – Could authors provide a magnification for Supplementary Figure 1C.

We have magnified the Supplementary Figure 1C as the reviewer requested.

Line 81 – It isn't clear what the different colors in the atlas picture represents in Figure 1D. Could the authors describe further what these colors mean? Additionally, authors should specify whether the reader is looking at a sagittal or coronal representation of the FN.

We have modified the text to specify that the different colors represent cell position from different animals analyzed, and that images correspond to coronal sections.

Line 183 – Figure legend for panel C is mislabeled.

We have corrected it.

Line 222 – It isn't entirely clear what the # symbols represent in Figure 5c and Figure 5d. The figure legend indicates that the # symbol is a difference between phases. The authors should specify which phases were different from which.

Lines have been added to the figure to clarify to which differences the symbols refer to.

Line 222 – Additionally, could the authors provide the behavior from the electrophysiological dataset. Prior experiments to this one in the study, CNO was given during EXT1 and EXT2 whereas in this experiment it was only given during EXT1. It would be interesting to further speculate on the relationship between spike burst occurrences during FN-MD inhibition and behavior.

We had indeed scored the immobility, but we chose not to include it in the MS because in these experiments, the animals in the conditioning box carry a wireless system and we observed that the animals expressed more immobility but had postures which differed much from the usual postures of freezing and seemed to be somewhat constrained by the wireless transmitter which was leaning against the floor/walls (see example Rev. Fig. 1 below). It is difficult to decide to which extent this corresponds to freezing or to resting.

Rev. Fig. 1: example of immobility posture observed during recordings: the wireless electrophysiological headstage is resting against the floor/wall. It is unclear to which extent this may correspond to freezing

The quantification of immobility (Rev. Fig. 2) suggests globally a lowering of immobility of the CT+SAL group in EXT3 consistent with the occurrence of extinction. Yet, the small number of mice (compared to behavioral groups) and the unreliability of the measure of immobility to assess freezing motivated our choice not to include the figure which sounded misleading. We have added a comment in the Methods to explain the difficulty to assess freezing during electrophysiological recordings.

Line 561- Reasons for performing the experience only in male mice should be specified.

Males and females mice are indeed equally suitable for fear conditioning analysis (Kaluve et al. 2022 Neurosci Biobehav Rev 143:104962). However, male mice are bigger and have therefore more strength to carry the implant and preamplifier in electrophysiological experiments. This is the reason of our preference and is now specified in the Methods.

Line 620 – Authors should explicitly specify how long tones were presented during extinction.

We have used 30 second-long CS with 30 sec inter-tone intervals(as in our previous study Frontera et al. 2020). This is now specified in the text.

Line 631 – It would help readers interpret anxiety-like behaviors if there was a timeline in which these anxiety-like behavioral tests were run. Were they done on consecutive days, all in one day?

Tests were performed on successive days one week before fear-conditioning experiments, starting with the less anxiogenic, Open Field, followed by Elevated plus maze and Dark-light box. Nociception was performed two weeks after the end of fear-conditioning experiments. This is now specified in the Methods.

Reviewer #2 (Remarks to the Author):

The contribution of the cerebellum to fear conditioning has long been ignored, in particular its contribution to extinction. Only recently, studies have started to investigate the contribution of the cerebellum to extinction of learned fear. This is the very first study investigating how the cerebellum may interact with the very well known cerebral fear extinction network. The study is carefully performed, shows the anatomical connections between fastigial nuclei, thalamus (MD) and prefrontal cortex (dmPFC); and shows that inhibition of this cerebellar-cerebral efferent pathway

impedes extinction. Different to what one may expect at first sight (the prelimbic cortex is generally known to be involved in the acquisition whereas the infralimbic cortex is known to be involved in extinction of learned fear responses), the authors show that inhibition of cerebellar output via the thalamus to prelimbic areas impedes extinction. Therefore, findings show that the cerebellum may contribute to extinction primarily via modulation of prelimbic areas. Authors also show how this may happen by demonstrating corresponding changes of thalamic and prefrontal cortex oscillations. The study is innovative (first of its kind) and of major importance for the field.

We thank the reviewer for these positive comments.

As stated in the acknowledgments of the paper, the reviewer has seen and commented upon a previous version of the manuscript. I have therefore few comments.

i) It reads in the manuscript as if anterior cingulate cortex and prelimbic cortex are different areas of the dmPFC. Is this correct, or does ACC refer to the human brain and prelimbic cortex to the rodent brain?

Anterior cingulate cortex and prelimbic cortex refer to different anatomical areas within the dmPFC of rodents, but the exact definition of the compartments of the mouse prefrontal cortex is still a topic of debates with some author suggesting that the PL should be splitted in two compartments belonging respectively to ventral and dorsal mPFC (e.g. Le Merre et al. 2021 Neuron 109(12):1925-1944). Since previous studies have demonstrated that dmPFC (including ACC and PL) projects to the lateral MD and has a functional role in fear extinction (Karalis et al. 2016 Nat Neurosci. 19(4): 605–612; Kloet et al. 2021 Nat Commun. 12(1):1994), we considered the dmPFC as the functional area involved in our study. Nevertheless, our electrophysiological sampling was mostly targeted at the PL (suppl. Fig 6).

ii) Out of curiosity: Did the authors investigate whether this cerebellar-thalamic-prefrontal cortex pathway impacts fear acquisition learning, as one would expect?

Unfortunately, this was out of the scope of the present study. Since the MD has long been described to have a role in extinction and not in fear learning we focused our research in the extinction process.

Dagmar Timmann

Reviewer #3 (Remarks to the Author):

In this report, Frontera et al. perform experiments designed to map and perturb a specific Fastigial nucleus to mediodorsal thalamus projection circuit during fear

extinction and also while making in vivo recordings from dorsomedial prefrontal cortex (dmPFC) and mediodorsal (MD) thalamus, two areas with reciprocal neuronal connections. They use rigorous mapping experiments with anterograde and retrograde approaches and find very specific circuit connections between caudal FN neurons to specific regions of the MD (contralateral lateral MD) enriched in neurons that are positive for calbindin expression. Next, they perform circuit perturbations with excitatory optogenetics, which induces responses in the MD and dmPFC, consistent with both direct and indirect pathways. These are again quite rigorous, with anterograde and retrograde viral approaches to elicit channelrhodopsin expression, though the authors report that there are differences in results with these two approaches (Figure 3). These experiments show that the FN circuit they are interested in is sufficient to elicit changes in firing in the MD and dmPFC. The authors then use inhibitory chemogenetics to test the role of these neurons in fear extinction and show that inhibition of this circuit increases freezing in their fear extinction paradigm (blocking the learning of new memory). They show that this manipulation does not alter baseline anxiety, or pain responses. Next, the authors show that chemogenetic inhibition in the FN->MD circuit increases burst firing in the MD. In a real strength of this paper, over the last 3 figures, the authors look at how the FN-MD circuit modulates reciprocal interactions between MD and dmPFC. They confirm that dmPFC shows 4Hz oscillations associated with fear extinction, and that this is correlated with a reduction in MD burst firing, which is reduced when the FN-MD circuit is chemogenetically inhibited in Figure 6. In fig 7, they show that phase locking of the MD bursting and dmPFC 4 Hz LFP oscillation is maintained when the FN-MD circuit is inhibited. In figure 8, they look at the coherence of 4Hz LFP in both MD and dmPFC and show that this is increased by chemogenetic inhibition of the FN-MD circuit.

Strengths:

This is an important, novel and rigorous contribution to the literature, demonstrating a role for a specific FN->MD circuit that is 1) necessary for fear extinction and 2) demonstrates mechanistically, that this circuit modulates reciprocal interactions between MD and dmPFC at the level of both single units and LFPs. This is a major and important contribution to the fear/response to threat literature, as well as the rapidly developing field in cerebellar-mediated cognitive/emotional functions. The data strongly supports the conclusions of the investigators, and I do not detect any major weaknesses in the work. The methodology is sound and is clearly presented.

We thank the reviewer for these positive comments.

Minor issues:

The authors have been very careful in controlling for different behaviors that may confound the interpretation of the results.

The one control I don't see is whether the inhibition of the FN-MD circuit alters sensorimotor gating at the level of acoustic startle or prepulse inhibition of the startle reflex—if there is a sensorimotor gating deficit, would this alter the interpretation of results?

We thank the reviewer for this question. Our data acquisition strategy was unfortunately not optimized to study startle (we used 25 Hz video rate, filmed from above, while a higher frame rate with side views would have been more optimal, see for example Pantoni et al. *Front Behav Neurosci.* 2020 14:83). Despite these limitations, we have estimated the motor reaction to CS onset following the methodology of Pantoni et al. by 1) assessing the rate of movement (measured by change in pixel values from successive frames); 2) computing the fractional change of the rate movement before and after CS onset. While the rate of change in pixel values was typically increased by several folds following immediately the onset of CS consistent with motor response to this onset, we did not observe an overt deficit of reactivity to the CS which would suggest a strong change in auditory gating. We feel however that this analysis is a bit too coarse to add this result to the MS.

Estimation of the movement induced by the CS onset for the 5 first CS presentation at the beginning of EXT1 for the main behavioral groups. No significant difference was found between the groups (Mann Whitney tests CT+SAL vs Gi+CNO: $p=0.25$, SHAM+SAL vs SHAM+CNO: $p=0.26$)

Methods: The startle reflex of the mice induced by the presentation of the CS was computed using an algorithm programmed in Python 3.6 with the OpenCV 4 library. Each frame obtained from the videos were analyzed according to the following process: the frame was transformed into a grayscale image. The grayscale frame was filtered using a bilateral threshold ($sd=9$, kernel 75 pixels by 75 pixels), then binarized using the Otsu algorithm implemented in OpenCV in order to differentiate the mouse from the background. Two consecutive frames were then subtracted, yielding a number of pixels changed from one frame to the other, reflecting the locomotor activity of the mouse. Then, the average pixel change during the 400 ms following the onset of the CS was divided by the average pixel change during the 400 ms preceding the onset of the CS, representing a fractional increase in motion induced by the presentation of the CS. Finally, the median ratio of pixel change for the first 5 CSs of Extinction 1 was reported for each mouse.

I'm curious if authors think that the role that the FN->MD circuit fits into a model of how the cerebellum affects behavior? The data strikes me as similar as what one might see in slower cerebellar adaptations to say, the vestibulo-ocular reflex or prism adaptation, as opposed to the rapid corrections it can elicit in other movements.

This is indeed a hard question (notably since the current knowledge of the link between oscillations and extinction is incomplete and could not be solved in the context of the present study). In the cases cited by the reviewer, the cerebellum is

likely the main site of learning, while in our case the extinction learning most likely reside in the dmPFC/amygdala network (as the current literature indicates). At this stage our speculation is that during extinction, there is a continuous 'choice' as to remain in the freezing state (linked to 4Hz) or to switch to a more active state (theta-band oscillations) and this 'choice' may impact the rate of extinction –see e.g. Ozawa, M. (2020). Nature Communications, 11(1), 1–16. Our data show that the cerebellum-MD pathway regulates the 4Hz oscillation (notably by controlling the rate of bursts which are typically followed by stronger 4Hz oscillations, see Fig 7h-k). Therefore the interpretation of our data would point toward the cerebellum regulating dmPFC oscillations/state, which would determine the rate of learning. Yet, the understanding of the link between dmPFC oscillations and extinction is still a debated topic, so the interpretation outlined above should be rated as quite speculative at this stage.

A corollary to this question is whether this circuit plays a role in associative fear learning.

This is indeed an interesting question (also raised by the reviewer 2), which was not investigated in the course of this study because of the lack of clear involvement of the dmPFC in fear learning.

The authors tell us that in Figure 4f, baseline freezing is not different between groups, though I'm very curious about Context A...it looks like there might be an effect of an outlier, which raises concern that the level of baseline freezing is possibly different between groups.

We have checked whether the removal of the outlier affected the result but it was not sufficient to reach significance ($p=0.089$, Wilcoxon's test). (NB: this figure panel has been expanded to show the context B data for all days on request of reviewer 1).

What is the interpretation of why the MD bursting is higher at baseline in the Gi+CNO mice in Fig 5?

Our results show that less cerebellar input induce increased MD bursting; this may reflect a predisposition to express freezing but it would not be triggered until dmPFC send the signal at the CS presentation. Actually, there is no proof of direct induction (sufficiency) of freezing by bursting in MD, rather it seems to limit extinction rate (Lee et al. Nature Neuroscience 15, 308-314 (2012)). Previous works have indicated that dmPFC 4 hz oscillation correlates with the freezing behavior, as we have discussed in our study. We have made some modifications in the text to clarify this point.

How were the optogenetic stimulation parameters selected?

We used the same parameters as established in our previous work (Frontera et al. 2020) based on ramps of intensities. This is now mentioned in the Methods section.

REVIEWER COMMENTS

Reviewer #1 (Remarks to the Author):

The authors have performed additional analyses and experiments to address the reviewer's concerns. The outcome of the new analyses, now shown in Figure 4, is a peculiar affect of chemogenetic inhibition of fastigial nucleus (FN) projections to the mediodorsal (MD) thalamus. In essence, only half of the mice showed an extinction impairment after CNO treatment; animals that showed higher EXT1 freezing showed the effect relative to animals that showed lower freezing. But overall the levels of freezing in the control animals across this median split were not incredibly different, so it is unclear why the chemogenetic manipulation was effective in only half of the animals. The paper is build around the mechanism by which FN->MD projections regulate extinction, but only a subset of the animals showed a behavioral affect of manipulating the pathway. It is not clear how the mechanistic investigation of this pathway proceeds when it is unclear what accounts for the differential sensitivity of extinction learning to chemogenetic manipulation of the projection. At minimum, the outcome of the work must be qualified to indicate that this projection is only engaged to mediate extinction in a subset of animals.

Reviewer #2 (Remarks to the Author):

All my questions have been answered.

Reviewer #3 (Remarks to the Author):

The authors have responded to the reviews satisfactorily. As mentioned previously, this is an important, novel and rigorous contribution to the literature, demonstrating a role for a specific FN->MD circuit that is 1) necessary for fear extinction and 2) demonstrates mechanistically, that this circuit modulates reciprocal interactions between MD and dmPFC at the level of both single units and LFPs. This is a major and important contribution to the fear/response to threat literature, as well as the rapidly developing field in cerebellar-mediated cognitive/emotional functions. The data strongly supports the conclusions of the investigators, and I do not detect any major weaknesses in the work. The methodology is sound and is clearly presented.

REVIEWER COMMENTS

Reviewer #1 (Remarks to the Author):

The authors have performed additional analyses and experiments to address the reviewer's concerns. The outcome of the new analyses, now shown in Figure 4, is a peculiar affect of chemogenetic inhibition of fastigial nucleus (FN) projections to the mediodorsal (MD) thalamus. In essence, only half of the mice showed an extinction impairment after CNO treatment; animals that showed higher EXT1 freezing showed the effect relative to animals that showed lower freezing. But overall the levels of freezing in the control animals across this median split were not incredibly different, so it is unclear why the chemogenetic manipulation was effective in only half of the animals. The paper is built around the mechanism by which FN->MD projections regulate extinction, but only a subset of the animals showed a behavioral affect of manipulating the pathway. It is not clear how the mechanistic investigation of this pathway proceeds when it is unclear what accounts for the differential sensitivity of extinction learning to chemogenetic manipulation of the projection. At minimum, the outcome of the work must be qualified to indicate that this projection is only engaged to mediate extinction in a subset of animals.

We have indeed clarified in the abstract and in the discussion that the impact of FN-MD inhibition is seen in mice with high initial freezing levels. Our results do not indeed fully resolve the mechanisms of action of the cerebellum in the regulation of fear extinction learning via the mediodorsal thalamus (MD). Increased MD bursting has been previously described to reduce fear extinction [Fig 6c of Lee et al. 2012] in protocols where mice started at high freezing levels (average ~ 80 % at the onset of extinction) similar to our “ high-freezing” group of mice (~ 80% at beginning of Extinction 1) while our “ low-freezing” mice start at ~ 60%. Our data could thus result from a non-linear sensitivity of the rate of extinction to MD bursting. We observed that MD bursting during 4Hz episodes is associated with a consecutive maintenance of the 4Hz-oscillatory state in the MD-dmPFC network (Fig 8h-k); this may indeed be part of the mechanism by which the MD prevents the mPFC to engage fear extinction learning by maintaining it in a “ fear” state ; the prevention of extinction would therefore be effective as long as fear expression is high enough. This remains speculative and has been added to the discussion as a possible link between our behavioral and mechanistic results.

References

1. Lee, S., Ahmed, T., Lee, S., Kim, H., Choi, S., Kim, D. S., Kim, S. J., Cho, J., & Shin, H. S. (2012). Bidirectional modulation of fear extinction by mediodorsal thalamic firing in mice. *Nature Neuroscience*, 15(2), 308–314. <https://doi.org/10.1038/nn.2999>

Reviewer #2 (Remarks to the Author):

All my questions have been answered.

Reviewer #3 (Remarks to the Author):

The authors have responded to the reviews satisfactorily. As mentioned previously, this is an important, novel and rigorous contribution to the literature, demonstrating a role for a specific FN->MD circuit that is 1) necessary for fear extinction and 2) demonstrates mechanistically, that this circuit modulates reciprocal interactions between MD and dmPFC at the level of both single units and LFPs. This is a major and important contribution to the fear/response to threat literature, as well as the rapidly developing field in cerebellar-mediated cognitive/emotional functions. The data strongly supports the conclusions of the investigators, and I do not detect any major weaknesses in the work. The methodology is sound and is clearly presented.

We thank the reviewer for these positive comments.